# DEUP: DIRECT EPISTEMIC UNCERTAINTY PREDICTION

## ABSTRACT

Epistemic uncertainty is the part of out-of-sample prediction error due to the lack of knowledge of the learner. Whereas previous work was focusing on model variance, we propose a principled approach for directly estimating epistemic uncertainty by learning to predict generalization error and subtracting an estimate of aleatoric uncertainty, i.e., intrinsic unpredictability. This estimator of epistemic uncertainty includes the effect of model bias (or misspecification) and is useful in interactive learning environments arising in active learning or reinforcement learning. In addition to discussing these properties of Direct Epistemic Uncertainty Prediction (DEUP), we illustrate its advantage against existing methods for uncertainty estimation on downstream tasks including sequential model optimization and reinforcement learning. We also evaluate the quality of uncertainty estimates from DEUP for probabilistic classification of images and for estimating uncertainty about synergistic drug combinations.

## 1 INTRODUCTION

A remaining great challenge for machine learning research is purposeful knowledge-seeking by learning agents, which can benefit from estimation of epistemic uncertainty, i.e., a measure of lack of knowledge, which could potentially be eliminated with enough data if the learner converges to a Bayes-optimal predictor. Epistemic uncertainty estimation is already a key ingredient in active learning and Bayesian optimization (Aggarwal et al., 2014; Frazier, 2018) as well as exploration in Reinforcement Learning (Kocsis & Szepesvári, 2006; Tang et al., 2017). Epistemic uncertainty (EU) thus tells us how much could be gained from learning around a particular area of state-space or input data space. But how should we even quantify it? Much previous work has focused on model variance (Gal & Ghahramani, 2016; Lakshminarayanan et al., 2017) as a proxy thereof, i.e., how different are the functions compatible with the learner's preferences and the data. However, this approach does not take into account model bias (or misspecification), due to the learner's finite capacity or preferences, self-imposed to avoid overfitting or reduce computational costs. Recent work (Masegosa, 2020) has also shown that Bayesian methods are suboptimal for learning predictive models when the model is misspecified. Fig. 1 illustrates with Gaussian Processes (GP) that a model's posterior variance is not necessarily a good predictor of the reducible generalization error.

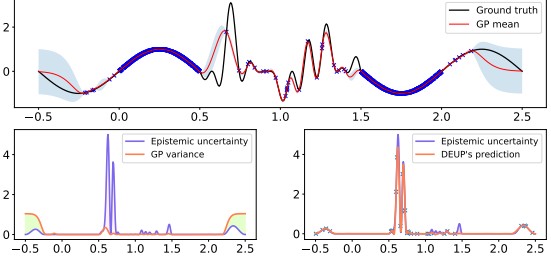

Figure 1: *Top.* A GP is trained on the blue points. GP uncertainty (model standard deviation) in shaded blue. *Left.* Using GP variance as a proxy for EU here misses out on important regions of the input space from which useful information can be acquired. *Right.* With extra out-of-sample points, DEUP learns an estimator of the EU that captures that information. Using this estimator to make decisions (e.g. active learning) would focus on where more data could most reduce the loss. The experiments show realistic use cases of DEUP in which less (or no) additional data is required to obtain reasonable uncertainty estimators.

Motivated by interactive learning settings arising for example in active learning, black-box optimization and reinforcement learning, we study as an alternative or complement the quantification of epistemic uncertainty in terms of the loss function. The total expected out-of-sample loss at a point can be decomposed into an epistemic (reducible) part and an aleatoric (irreducible) part. Aleatoric uncertainty is output noise, the part of the error that is intrinsic to the distribution of interest (and independent of the choice of the

learner), and it is also the error which may be achieved by a Bayes-optimal predictor. Consider the case of a learning system with universal approximation properties, like modern deep learning, whose effective capacity can be increased with more data, either via model selection, early stopping or other means. In such cases, the part of the error which is not aleatoric is reducible with more data. For this reason, we propose to define epistemic uncertainty as the difference between the expected generalization error and aleatoric uncertainty. Estimating it is useful to an active or interactive learner because it could be reduced with enough data, especially in regions of the input space we care about, i.e., where EU is large. In order to estimate EU, we propose DEUP, for Direct Epistemic Uncertainty Prediction, where we train a side learner (the uncertainty predictor) with an appropriate objective and training data in order to predict this generalization error. An estimate of EU can thus be obtained by subtracting an estimate of aleatoric uncertainty.

In this paper we thus focus on two predictors: the *main predictor* (for the learning task of original interest) and an *error predictor* which can be used to predict epistemic uncertainty of the main predictor at given points, given some knowledge or estimate of the aleatoric uncertainty. The proposed methodology can be seen as an end-to-end approach to epistemic uncertainty estimation, i.e., to help estimate where a learner would acquire the most information in terms of out-of-sample loss reduction, without relying on the brittle relationship between model variance and generalization error, which can get murky in the presence of misspecification (bias), as we argue below. In order to select points with high EU, one would still need some kind of search (in the space of $x$), e.g., to propose new candidates for active learning (Aggarwal et al., 2014; Nguyen et al., 2019; Bengio et al., 2021) or sequential model optimization (Kushner, 1964; Jones et al., 1998; Snoek et al., 2012), or exploration in RL (Kocsis & Szepesvári, 2006; Osband et al., 2016; Janz et al., 2019).

An interesting potential advantage of the proposed approach, compared with model variance estimators, is that DEUP error predictors can be explicitly trained to care about, and calibrate for estimating the generalization error for examples which may come from a distribution different from the distribution of most of the training examples, i.e., an out-of-distribution (OOD) setting. Such non-stationarities arise naturally in contexts like active learning and reinforcement learning (RL) because the learner explores new areas of the input space. In these non-stationary settings, we typically want to retrain the main predictor as we acquire new training data, not just because more training data is generally better but also to better track the changing distribution and generalize to yet unseen but upcoming OOD inputs. This setting makes it challenging to train the main predictor but it also entails an even greater non-stationarity for the training data seen by the error predictor: a large error initially made at a point $x$ before $x$ is incorporated in the training set (along with an outcome $y$) will typically be greatly reduced after updating the main predictor with $(x, y)$. We propose several strategies to cope with this non-stationarity, the most important being to rely on additional features (such as log-density estimates and model variance estimates at $x$). These features account for variations in prediction error arising due to updates of the main predictor, making DEUP's training distribution more stationary.

The experiments presented here explore first the scenario of purely trying to estimate epistemic uncertainty in terms of the recoverable generalization error, and second how this can be used in contexts of sequential model optimization (or black-box optimization) and exploration in RL (where the issue of non-stationarity arises). The main contributions of this paper are thus the following:

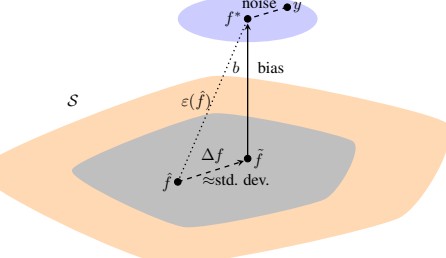

Figure 2: *Illustrating noise and bias. Observed $y$ is independent noise plus true $E[Y|x] = f^*(x)$, itself best approximated by unknown $\tilde{f}(x)$ in parametric set $S$ (orange), e.g., using Bayesian posterior distribution (grey) over parameters. $\tilde{f}$ is the closest function in $S$ to $f^*$, leading to a bias $b(x) = f^*(x) - \tilde{f}(x)$. $\varepsilon(\hat{f})(x) = f^*(x) - \hat{f}(x)$ is the reducible error of the main predictor $\hat{f}$ (e.g. posterior mean), whose square corresponds to the epistemic uncertainty or lack of knowledge, that DEUP aims at estimating. With $\tilde{f}$ the unknown ideal predictor and $\hat{f}$ the actual (e.g. mean) predictor, the square of $\Delta f(x) = \tilde{f}(x) - \hat{f}(x)$ induces variance in the posterior. $\varepsilon(\hat{f})(x) = b(x) + \Delta f(x)$ indicates that using the variance as a proxy for lack of knowledge misses out a non-negligible quantity: the bias $b(x)$.*

1. Novel end-to-end approach for directly estimating the epistemic uncertainty as the lack of knowledge about the true $f^*(x) = E[Y|x]$, the Direct Epistemic Uncertainty Prediction (DEUP) approach.

2. Experimental evidence that this direct estimation of epistemic uncertainty can lead to more precise estimates of the lack of knowledge than variance-based methods.

3. A mean to mitigate the issue of non-stationarity arising in training the uncertainty estimator in interactive settings.

4. A successful validation of the above in the contexts of (a) sequential model optimization and (b) exploration in RL.

## 2   ALEATORIC AND EPISTEMIC UNCERTAINTIES, MODEL VARIANCE

Consider a supervised learning algorithm (or learner) $\mathcal{L}$ mapping a dataset $\mathcal{D}$ to a predictive function $\hat{f} = \mathcal{L}(\mathcal{D})$. $\mathcal{L}$ tries to minimize the expected value of a supervised learning loss $l(\hat{f}(x), y) \in \mathbb{R}$ under unknown $P(Y|x)$. In this section, we focus on regression tasks with the squared loss $l(\hat{y}, y) = (\hat{y} - y)^2$, with $y \in \mathbb{R}$. In Appendix B, we provide the corresponding results for a general loss function.

**Definition 2.1** *The **expected loss** of a predictor $\hat{f}$ at $x$ is defined as:*

$$\mathcal{U}(\hat{f}, x) = \int (\hat{f}(x) - y)^2 dP(y|x). \tag{1}$$

Note how the expected loss is an unknown quantity, because we generally do not have access to the true data generating distribution $P(Y|x)$. The errors made by any predictor $\hat{f}$ at $x$ are due to both the inherent randomness of $P(Y|x)$ (aleatoric uncertainty) and the lack of knowledge of the predictor that can be tackled by acquiring more information around $x$ (epistemic uncertainty). Next, we will define these concepts more formally. Because of this natural decomposition of the expected loss, we will also refer to it as **total uncertainty**, and we will use the two terms interchangeably.

Bayes-optimal predictors $f^*$ satisfy the following equation at every $x$:

$$\forall \tilde{y} \in \mathbb{R} \quad \int (f^*(x) - y)^2 dP(y|x) \leq \int (\tilde{y} - y)^2 dP(y|x).$$

They depend on the underlying data distribution only and not on learner $\mathcal{L}$ or trained predictor $\hat{f}$. Additionally, all Bayes-optimal predictors have the same total uncertainty for all $x$. Aleatoric uncertainty is the irreducible expected error, i.e., that made by a Bayes-optimal predictor:

**Definition 2.2** *We define **aleatoric uncertainty** at $x$ as the total uncertainty of any Bayes-optimal predictor $f^*$ at $x$:*

$$\mathcal{A}(x) = \mathcal{U}(f^*, x). \tag{2}$$

*and we note that by definition $\mathcal{A}(x) \leq \mathcal{U}(f, x), \ \forall f \ \forall x$.*

We now propose a definition of epistemic uncertainty of predictor $\hat{f}$ as the gap between the error of $\hat{f}$ at $x$ and the lowest possible error at $x$, i.e., the reducible part of the loss, given more knowledge.

**Definition 2.3** *We define **epistemic uncertainty** $\mathcal{E}(\hat{f}, x)$ of a predictor $\hat{f}$ at $x$ as*

$$\mathcal{E}(\hat{f}, x) = \mathcal{U}(\hat{f}, x) - \mathcal{A}(x) = \mathcal{U}(\hat{f}, x) - \mathcal{U}(f^*, x). \tag{3}$$

Using these definitions, we can present the main result about the regression setting.

**Proposition 1** *In a regression task with ground truth $P(y|x) = \mathcal{N}(y; \ f^*(x), \ \sigma^2(x))$,*

$$\mathcal{E}(\hat{f}, x) = (\hat{f}(x) - f^*(x))^2$$
$$\mathcal{U}(\hat{f}, x) = E_{y \sim P(.|x)}[(\hat{f}(x) - y)^2] = (\hat{f}(x) - f^*(x))^2 + \sigma^2(x)$$
$$\mathcal{A}(x) = E_{y \sim P(.|x)}[(f^*(x) - y)^2] = \sigma^2(x)$$

The proof is given in Appendix C. Additionally, we present in Appendix B a similar result for the negative log-likelihood loss, with discrete or continuous outputs.

We will discuss in the remainder of this section the relation between this new definition of epistemic uncertainty and the more traditional way of modelling it through model variance or entropy.

Consider a parametric model $p(Y|x, \theta)$ and a learner maintaining a distribution over parameters $\theta \in \Theta$, each corresponding to a predictor $f$ in a parametric set of functions $S$, possibly starting from a prior $p(\theta)$ that would lead to a posterior distribution $p(\theta|\mathcal{D})$. Clearly, the fact that multiple $\theta$'s and corresponding values of $f$ are compatible with the data and the prior indicates lack of knowledge. Because the lack of knowledge indicates where an interactive learner should acquire more information, this justifies the usage of dispersion measures, such as the variance or the entropy of the posterior predictive, as proxies for epistemic uncertainty.

However, the limited capacity of $S$ or the prior $p(\theta)$ may pull the optimal $\tilde{f}$ in $S$ away from the Bayes-optimal predictor $f^*$. We can refer to these self-imposed constraints as a form of bias, in the sense that the learner is usually biased towards the prior preferences, e.g., towards smoother predictors. In particular, this issue can arise when training neural networks with limited data, where the network may not use all of its capacity due to implicit (and not fully understood) regularization properties of SGD (Kale et al., 2021), explicit regularization, early stopping or a combination of these, which can induce a preference on the functions it learns. In Fig. 2, we illustrate this gap with the bias function $b(x) = f^*(x) - \tilde{f}(x)$. Because of this bias, model variance cannot be an accurate measure of lack of knowledge $\mathcal{E}(\hat{f}, x) = (\hat{f}(x) - f^*(x))^2$ in the general case, as we have shown with the results in Fig. 1. In Deep Ensembles (Lakshminarayanan et al., 2017) e.g., if all the networks in the ensemble tend to fail in systematic ways, this aspect of prediction failure will not be captured by variance. Whereas Deep Ensembles variance provide us uncertainty regarding which of the networks we could draw is correct, this does not tell us how poor that network is even in a noise-free setting. On the other hand, with flexible models like neural networks, adding examples around $x$ where $b(x)^2$ is large may allow to increase capacity around $x$ and reduce $b(x)^2$.

In section 5, we will provide more experimental evidence supporting our claim that directly estimating the epistemic uncertainty as the reducible part of the loss can lead to better measures of lack of knowledge, compared to variance-based methods, especially in interactive learning settings.

## 3 DIRECT EPISTEMIC UNCERTAINTY PREDICTION

DEUP (Direct Epistemic Uncertainty Prediction) **uses observed out-of-sample errors in order to train an error predictor which can be used to estimate epistemic uncertainty** elsewhere, as suggested directly by definitions 2.1-2.3. These may be in-distribution or out-of-distribution errors, depending on whichever we care about. The pseudo-code of DEUP in interactive settings is provided below. The pseudo-code for the simpler version with a fixed training set is given in Appendix E.

### 3.1 FIXED TRAINING SET

Consider the simpler scenario with fixed training set $\mathcal{D}$ and learning algorithm $\mathcal{L}$ yielding predictor $\hat{f}$. We assume that $N$ out-of-sample (validation) points have been set aside and can thus be used for training an error predictor.

Training an *error* predictor $e$ with (input, target) pairs $(x, (\hat{f}(x) - y)^2)$ and squared loss yields an estimator of the total uncertainty. To see that this is the right training set for the secondary predictor $e$, we can assess what happens at the limit of infinite data: the estimator is asymptotically unbiased, i.e. $\lim_{N \to \infty} e(x) = \mathcal{U}(\hat{f}, x) = E[(\hat{f}(x) - Y)^2]$ for all $x$, if the learning algorithm ensures asymptotic convergence to a Bayes-optimal predictor. Examples of such learners include $k$-nearest neighbors with $k$ increasing at a proper rate slower than $n$, and neural networks whose size and regularization are hyper-parameters optimized on a growing validation set.

If $\mathcal{A}(x) = 0$ then $e(x)$ is an estimate of $\mathcal{E}(\hat{f}, x)$ as well as of $\mathcal{U}(\hat{f}, x)$. If an estimator $a(x)$ of aleatoric uncertainty is available, then $e(x) - a(x)$ becomes an estimate of the epistemic uncertainty of $\hat{f}$ at $x$. Alternatively, one could train $e$, constrained to be positive (e.g. with a softplus$(u) = \log(\exp(1 + u))$ output non-linearity), with examples $(x, (\hat{f}(x) - y)^2 - a(x))$ and also obtain an estimator of epistemic uncertainty. When we have access to an oracle that samples $Y$ given query $x$ from the environment

$P(Y|x)$ (e.g., in active learning or black-box optimization), then $\mathcal{A}(x)$ can be estimated using the empirical variance of different outcomes of the oracle at the same input $x$; see Appendix D.

Note that because $\mathcal{U}$ and $\mathcal{E}$ are functions of the predictor $\hat{f}$ as well (and $\hat{f}$ is itself derived from the training data $\mathcal{D}$), the error predictor can also be trained on features that are informative of the pair $(\hat{f}, x)$ rather than $x$ only. We will mainly use variance estimates (V), obtained from a separate model (e.g. a GP), and log-density estimates (D), in addition to the input $(x)$. The usage of these features is particularly important in non-stationary settings, as discussed next.

## 3.2 DEUP IN INTERACTIVE SETTINGS

Interactive settings, in which epistemic uncertainty estimation is used in order to guide information acquisition and iteratively gather new examples, provide a more interesting use case for DEUP (as shown in Figures 4 and 5, where good epistemic uncertainty estimation translates in faster learning curves). These settings are however more challenging: the main predictor is going to be retrained multiple times, with the new examples being added to the observed dataset each time. Two kinds of non-stationarities arise here:

First, the new examples are not drawn i.i.d. (since their choice depends on the epistemic uncertainty estimator obtained from earlier examples). This makes the stream of data seen by the main predictor non-stationary. For example in sequential model optimization or RL, the new examples may concentrate more and more towards regions where the reward might be higher. Whereas it is not clear how variance-based predictors can handle this, DEUP can in principle cope with it by virtue of having the error predictor trained not on out-of-sample and in-distribution examples but on out-of-sample and out-of-distribution examples of the kind we care about (e.g., as obtained from the interactive learning process). The error predictor then learns to predict the error which would be made on typical new examples coming from what we may call by analogy the "frontier of knowledge", while this frontier continuously expands due to exploration and interactive learning: we expect such out-of-distribution errors to be larger than in-distribution out-of-sample errors.

Second, the error predictor itself faces even more distributional change, even if the data stream seen by the main predictor was i.i.d.. Indeed, after example $x$ was selected, the main predictor will be retrained using that additional data point, and $x$ would now have much lower epistemic uncertainty. This means that the targets of the error predictor also change, as the main predictor is updated with newly acquired points.

Because the dataset $\mathcal{D}$ that is used to train the main predictor changes at each step of an active learning process, it is necessary to see $\mathcal{U}$ and $\mathcal{E}$ as functions of the dataset as well: $\mathcal{U}(\hat{f}, x) = \mathcal{U}(\mathcal{L}(\mathcal{D}), x)$, meaning that we can no longer use (input, output) pairs $(x, \hat{f}(x) - y)^2)$ as mentioned in section 3.1 to train an estimator of the total uncertainty, but should in principle use (input, output) pairs $((x, \mathcal{D}), (\hat{f}(x) - y)^2))$, where $\hat{f} = \mathcal{L}(\mathcal{D})$. However, $\mathcal{D}$ being a very high-dimensional object, with size growing with the number of acquired points, we may face severe overfitting issues when training an uncertainty estimator using $(x, \mathcal{D})$ as inputs. Hence, in this paper, we propose to use **stationarizing features** of the dataset $\mathcal{D}$ at $x$, that we denote by $\phi_{\mathcal{D}}(x)$, as inputs to the error predictor instead of $(x, \mathcal{D})$.

In addition to the input $x$ itself, we explored the use of two scalar features to stationarize the training set of the error predictor: a model variance estimate, and a log-density estimate. Both have been used as proxies for epistemic uncertainty in the literature. Furthermore, we considered a binary feature $s$ to incorporate in $\phi_{\mathcal{D}}(x)$, that specifies whether $x$ was already used to train the main predictor (or equivalently, $x \in \mathcal{D}$) or not.

More formally, we use $\phi_{\mathcal{D}}(x) = \left(x, s, \log \hat{q}(x|\mathcal{D}), \hat{V}(\tilde{\mathcal{L}}, \mathcal{D}, x)\right)$, where $\hat{q}(x|\mathcal{D})$ is a density estimate from data $\mathcal{D}$ at $x$, $s = 1$ if $x \in \mathcal{D}$ otherwise 0, $\tilde{\mathcal{L}}$ a learner that produces a distribution over predictors, which can chosen to be the same as $\mathcal{L}$, and $\hat{V}(\tilde{\mathcal{L}}, \mathcal{D}, x)$ is an estimate of the model variance of $\tilde{\mathcal{L}}$ trained on $\mathcal{D}$ at $x$. $\hat{q}$ is obtained by training a third model (besides the main predictor and the error predictor) as a density estimator (such as a Kernel Density Estimator or a flow-based deep network (Rezende & Mohamed, 2015), in our case). Like the other predictors, the density estimator also needs to be fine-tuned when new data is added to the training set.

In our experiments, we found that using inputs $\phi_{\mathcal{D}}(x)$, or even a subset of the 4 possible features, is sufficient to train an uncertainty estimator with targets $(\hat{f}(x) - y)^2$.

The pseudo-code is provided in Algorithm 1. Note that $\mathcal{D}_e$ is incremented twice with $x_{acq}$ but $\phi_{\mathcal{D}}(x_{acq})$ is different each time because $x_{acq}$ first is and then is not yet in $\mathcal{D}$.

### 3.2.1 KICKSTARTING THE ERROR PREDICTOR

In practical applications, whether in a fixed dataset setting or in interactive settings, it is preferable to use all the available data to train the main predictor, rather than keeping a subset to train the error predictor. This would entail inaccurate uncertainty estimates once real out-of-sample unlabelled data comes in. To tackle this, we propose a scheme inspired by standard $K-$ fold cross-validation in order to kickstart the error predictor with the initially available data. Multiple versions of the main predictor, trained on different subsets of the training data $\mathcal{D}$, are used to define targets for the error predictor, whereas the subsets themselves provide the input features. The scheme is summarized in the Optional Cross-Validation step of Algorithm 1. Note that in the case of multi-class classification tasks (with $n$ classes), we obtain the folds by splitting the data based on classes, with each fold containing $\lfloor n/K \rfloor$ classes. So when we train on $K - 1$ folds, the $\lfloor n/K \rfloor$ classes from the remaining fold are *out-of-sample*.

---

**Algorithm 1** Training procedure for DEUP in an Active Learning setting

---

**Data:** $\mathcal{D}_{init}$ initial training dataset with pairs $(x, y) \in \mathcal{X} \times \mathbb{R}$
$a : \mathcal{X} \mapsto \mathbb{R}$, estimator of aleatoric uncertainty $\quad \hat{f} : \mathcal{X} \mapsto \mathbb{R}$, main predictor (of $y$ given $x$)
$e : \mathcal{X} \mapsto \mathbb{R}$, total uncertainty estimator $\quad \phi : \mathcal{X} \mapsto \mathbb{R}^k$: chosen stationarizing features
$acq$: acquisition machinery that proposes new input points $x_a cq$ from $\mathcal{X}$, using the current $\hat{f}$ and $e$
**Training:**
Initialize empty dataset to train the uncertainty estimator, $\mathcal{D}_e$
Initialize $\mathcal{D} = \mathcal{D}_{init}$, the dataset of training points seen so far
**if** *kickstart_error_predictor* **then**
    **Cross-validation step:** Pre-fill $\mathcal{D}_e$ using training errors on initial training data $\mathcal{D}_{init}$:
    **while** *stopping criterion not reached* **do**
        Split $\mathcal{D}$ into $K$ random subsets $\mathcal{D}_1, \ldots, \mathcal{D}_K$ of equal size. Define $\tilde{\mathcal{D}} = \bigcup_{k=1}^{K-1} \mathcal{D}_k$
        Fit $\hat{f}$ on $\tilde{\mathcal{D}}$, and fit the features $\phi_{\tilde{\mathcal{D}}}$ on $\tilde{\mathcal{D}}$
        $\mathcal{D}_e \leftarrow \mathcal{D}_e \cup \bigcup_{(x,y) \in \mathcal{D}} \left\{ \left( \phi_{\tilde{\mathcal{D}}}(x), (y - \hat{f}(x))^2 \right) \right\}$
        Fit $e$ on $\mathcal{D}_e$
    **end**
**end**
$x_{acq} \leftarrow \varnothing, y_{acq} \leftarrow \varnothing$
**while** *stopping criterion not reached* **do**
    **Optional (or every few iterations only):** Fit $a$ on $\mathcal{D}$ (e.g. see Appendix D)
    Fit $\hat{f}$ on $\mathcal{D}$
    Fit features in $\phi_{\mathcal{D}}$ on $\mathcal{D}$ (e.g. density estimation)
    $\mathcal{D}_e \leftarrow \mathcal{D}_e \cup \left\{ \left( \phi_{\mathcal{D}}(x_{acq}), (y_{acq} - \hat{f}(x_{acq})^2 \right) \right\} \quad$ **if** $x_{acq} \neq \varnothing$
    Fit $e$ on $\mathcal{D}_e$
    $x_{acq} \leftarrow acq(\mathcal{X}, \hat{f}, e - a)$ (can be either a single point, or a batch of points)
    Sample outcomes from the ground truth distribution: $y_{acq} \sim P( . | x_{acq})$
    $\mathcal{D}_e \leftarrow \mathcal{D}_e \cup \left\{ \left( \phi_{\mathcal{D}}(x_{acq}), (y_{acq} - \hat{f}(x_{acq})^2 \right) \right\}$
    $\mathcal{D} \leftarrow \mathcal{D} \cup \{(x_{acq}, y_{acq})\}$
**end**

---

## 4 RELATED WORK

Kiureghian & Ditlevsen (2009) characterized the sources of uncertainty as aleatoric (inherent noise) and epistemic (incomplete knowledge). Gaussian Processes or GPs (Williams & Rasmussen, 1995), provide a way to capture epistemic uncertainty through the disagreement between the different predictors that fit the data.

In a deep learning context, Blundell et al. (2015); Kendall & Gal (2017); Depeweg et al. (2018) use the posterior distribution of network weights (MacKay, 1992) in Bayesian Neural Networks (BNNs) to capture epistemic uncertainty. Other techniques that rely on measuring the discrepancy between different predictors as a proxy for epistemic uncertainty include MC-Dropout (Gal & Ghahramani,

2016), that interprets Dropout (Hinton et al., 2012) as a variational inference technique in BNNs. These approaches, because they rely on sampling multiple sets of weights or dropout masks at inference time, share some similarities with ensemble-based methods, that include bagging (Breiman, 1996) and boosting (Efron & Tibshirani, 1994), in which multiple predictors are trained, and used jointly to make a prediction, although the latter measure variability due to the training set instead of the spread of functions compatible with the given training set, as in Bayesian approaches. For example, Shaker & Hüllermeier (2020) use Random Forests (Breiman, 2001) to estimate epistemic uncertainty. Deep Ensembles (Lakshminarayanan et al., 2017) are closer to the Bayesian approach, using an ensemble of neural networks that differ because of randomness in initialization and training (as you would have with MCMC, albeit in a more heuristic way). In addition to this, several other variants of this central idea of measuring discrepancy between different predictors have been proposed recently (Malinin & Gales, 2018; Tagasovska & Lopez-Paz, 2019; Amini et al., 2020; Liu et al., 2020; van Amersfoort et al., 2020; Wen et al., 2020; Antoran et al., 2020; Kirichenko et al., 2020; Malinin et al., 2020a; van Amersfoort et al., 2021). We discuss these methods in more detail in Appendix A.

More closely related to DEUP, Yoo & Kweon (2019) proposed a loss prediction module for learning to predict the value of loss function. Hu et al. (2020) also propose using a separate network that learns to predict the variance of an ensemble. These methods, however, are trained only to capture the in-sample error, and do not capture the out-of-sample error which is more relevant for scenarios like active learning where we want to pick $x$ where the reducible generalization error is large. EpiOut (Umlauft et al., 2020; Hafner et al., 2019) propose learning a binary output that simply distinguishes between low or high epistemic uncertainty.

## 5 EXPERIMENTS

### 5.1 UNCERTAINTY ESTIMATION

#### 5.1.1 IMPORTANCE OF EPISTEMIC UNCERTAINTY CALIBRATION

In tasks that use uncertainty estimates to make decisions, and when the resources are limited, it is important to have well calibrated uncertainty estimates, to avoid exploring uninteresting areas of the search space, as we will observe in the following subsections. We aim at comparing DEUP, which presumably takes into account both bias and variance, with established methods that are based on variance. Hence, we design a function with varying degrees of smoothness (as measured by the absolute values of its derivatives), and use fewer training points in the less smooth regions, where epistemic uncertainty will thus be higher. We first consider the scenario described in Section 3.1, where the task is to regress a known one-dimensional function.

In Figure 1, we use a GP as a predictor, and compare the GP's variance to the epistemic uncertainty as per Eq. 3 (the square loss between the GP's predicted mean and the ground truth, as per Proposition 1). The figure shows that, naturally, in regions of the input space where little or no data is available, and especially in the less smooth regions, the predicted variance underestimates the epistemic uncertainty (squared error) by a higher margin compared to the other regions. We see that training another GP to directly estimate this squared loss, using a small held out out-of-sample data set for which the ground truth value is known, allows to close the gap between the predicted uncertainty and the true epistemic uncertainty. Note that this second predictor can be trained using any chosen learning algorithm $\mathcal{L}$.

Naturally, if we had access to a held out set, it would be wiser to use it as part of the training set of the main predictor, rather than the uncertainty predictor. Alternatively, we could use *cross-validation* splits (Section 3.2.1), at the expense of computational efficiency, to avoid discarding precious data. In active learning, it is the acquired points, before they are used to retrain the main predictor, that also act as the *out-of-sample* examples to train DEUP. Using the stationarizing features introduced in Section 3.2, these acquired points should be informative enough for DEUP to generalize its uncertainty estimates to unknown regions of the search space. In RL, because the targets (e.g. of Q-Learning) are themselves estimates and moving, data seen at any particular point is normally out-of-sample and can inform the uncertainty estimator, when the inputs are used with the stationarizing features.

#### 5.1.2 EPISTEMIC UNCERTAINTY PREDICTIONS FOR REJECTING DIFFICULT EXAMPLES

Epistemic uncertainty estimates can be used to reject difficult examples where the predictor might fail, such as OOD inputs[1]. We thus consider a standard OOD Detection task (van Amersfoort et al., 2020; 2021), where we train a ResNet-18 (He et al., 2016) for CIFAR-10 classification (Krizhevsky, 2009)

---

[1]e.g. rare but challenging inputs can be directed to a human, avoiding a costly mistake

Table 1: *Spearman Rank Correlation Coefficient (SRCC) between predicted uncertainty and OOD generalization error (SVHN); Area under ROC Curve (AUROC) for OOD Detection (SVHN) with CIFAR-10 ResNet-18 models (3 seeds). DEUP significantly outperforms baselines.*

| Model | SRCC | AUROC |
|---|---|---|
| MC-Dropout Gal & Ghahramani (2016) | $0.287 \pm 0.002$ | $0.894 \pm 0.008$ |
| Deep Ensemble Lakshminarayanan et al. (2017) | $0.381 \pm 0.004$ | $0.933 \pm 0.008$ |
| DUQ van Amersfoort et al. (2020) | $0.376 \pm 0.003$ | $0.927 \pm 0.013$ |
| DUE van Amersfoort et al. (2021) | $0.378 \pm 0.004$ | $0.929 \pm 0.005$ |
| DEUP (D+V) | $\mathbf{0.426 \pm 0.009}$ | $0.933 \pm 0.010$ |

and reject OOD examples using estimated uncertainty in the prediction. To facilitate rejection of classes other than those in the training set, we use a Bernoulli Cross-Entropy Loss for each class (van Amersfoort et al., 2020): $l(\hat{f}(x), y) = -\sum_i y_i \log \hat{f}_i(x) + (1 - y_i) \log(1 - \hat{f}_i(x))$, where $y$ is a one-hot vector ($y_i = 1$ if $i$ is the correct class, and 0 otherwise), and $\hat{f}_i(x)$ = predicted probability for class $i$, so the target for out-of-distribution data (from other classes) is $y = \{0, \ldots, 0\}$. To ascertain how well an epistemic error estimate sorts non-training examples by the above NLL loss, we consider the rank correlation between the predicted uncertainty and the observed OOD generalization error on SVHN examples (Netzer et al., 2011). This metric focuses on the quality of the uncertainty estimates rather than just their ability to simply classify in- vs out-of-distribution. We also report the AUC for the OOD detection task; more details in Appendix F. Table 1 shows that with the variance from DUE (van Amersfoort et al., 2021) and the density from MAF (Papamakarios et al., 2017) as DEUP inputs, we obtain uncertainty estimates that have high rank correlation with the underlying generalization errors and competitive AUROC, compared with the baselines. In addition, since the error predictor is trained separately from the main predictor there is no explicit trade-off between the accuracy of the main predictor and the quality of uncertainty estimates. We achieve competitive accuracy of 93.89% for the main predictor. We ignore the effect of aleatoric uncertainty (due to inconsistent human labelling), which would require a human study to ascertain. We present additional results in a distribution shift setting in Appendix F.

### 5.1.3 EPISTEMIC UNCERTAINTY ESTIMATION FOR DRUG COMBINATIONS

We validate DEUP in a real-world regression task predicting the synergy of drug combinations. While much effort in drug discovery is spent on finding novel small molecules, a potentially cheaper method is identifying combinations of pre-existing drugs which are synergistic (i.e., work well together). However, every possible combination cannot be tested due to the high monetary cost and time required to run experiments. Therefore, developing good estimates of epistemic uncertainty can help practitioners select experiments that are both informative and promising. As shown in Table 3(c), the out-of-sample error predicted by DEUP correlates better with residuals of the model on the test set in comparison to several other uncertainty estimation methods. Moreover, DEUP better captured the order of magnitude of the residuals as shown in Figure 3. Details on experiments and metrics are in Appendix J.

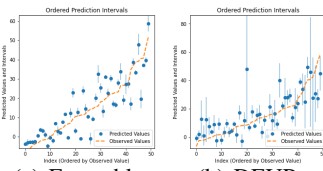

| Model | Corr. w. res. | U. Bound | Ratio | Log Likelihood |
|---|---|---|---|---|
| MC-Dropout | $0.14 \pm 0.07$ | $0.56 \pm 0.05$ | $0.25 \pm 0.12$ | $-20.1 \pm 6.8$ |
| Deep Ensemble | $0.30 \pm 0.09$ | $0.59 \pm 0.04$ | $0.50 \pm 0.13$ | $-14.3 \pm 4.7$ |
| DUE | $0.12 \pm 0.12$ | $0.15 \pm 0.03$ | $\mathbf{0.80 \pm 0.79}$ | $-13.0 \pm 0.52$ |
| DEUP | $\mathbf{0.47 \pm 0.03}$ | $0.63 \pm 0.05$ | $\mathbf{0.75 \pm 0.07}$ | $\mathbf{-3.5 \pm 0.25}$ |

(a) Ensemble     (b) DEUP

(c) Quality of uncertainty estimates from different methods.

Figure 3: Drug Combinations. Predicted mean and uncertainty on test set. 50 test examples are ordered by increasing value of true synergy score (orange). Model predictions and uncertainties in blue. Ensemble **(a)** (and MC-dropout, not shown) consistently underestimate uncertainty while DEUP **(b)** captures the right order of magnitude. **(c)** *Corr. w. res.* shows correlation between model residuals and predicted uncertainties $\hat{\sigma}$. A best-case *Upper Bound* on *Corr. w. res.* is obtained from the correlation between $\hat{\sigma}$ and true samples from $\mathcal{N}(0, \hat{\sigma})$. *Ratio* is the ratio between col. 1 and 2 (larger is better). *Log-likelihood*: average over 3 seeds of per sample predictive log-likelihood.

### 5.2 SEQUENTIAL MODEL OPTIMIZATION

Sequential model optimization or black-box optimization are forms of active learning. At each stage, the learner chooses query examples to label, looking for examples with a high value of the unknown oracle function. Such examples are selected so they have a high predicted value (to maximize the unknown oracle function) and a large predicted uncertainty (offering the opportunity of discovering yet higher values). Acquisition functions, such as Upper Confidence Bound (UCB, Srinivas et al. (2010)) and Expected Improvement (EI, Močkus (1975)) trade-off exploration and exploitation, and

one can select the next candidate by looking for $x$'s maximizing the acquisition function. We combine UCB and EI with DEUP (DEUP-UCB and DEUP-EI) to perform active learning, treating the main predictor and DEUP epistemic uncertainty prediction at $x$ respectively as mean and variance of a Gaussian distribution for the learner's guess of the value of the oracle at $x$. We showcase how using DEUP to calibrate GP variances (used as input for DEUP) allows for better performances in higher-dimensional optimization tasks. Specifically, we compare DEUP-EI to TuRBO-EI (Eriksson et al., 2019), a state-of-the-art method for sequential optimization, that fits a collection of local GP models instead of a global one in order to perform efficient high-dimensional optimization, on the Ackley function (Ackley, 2012) as oracle, a common benchmark for optimization algorithms. The oracle function can be defined for arbitrary dimensions, and has many local minima.

In Figure 4, we compare different methods on the Ackley-10 function, in addition to the optimum reached in budget-constrained optimization problems for different oracle input dimensions, and we find that adapting DEUP to TuRBO consistently outperforms regular TuRBO, especially in higher dimensions. See also Appendix H for 1D and 2D black-box optimization tasks where DEUP-EI outperforms GP-EI (Bull, 2011), as well as neural networks with MC-Dropout or Ensembles. We find GP-EI getting stuck in local optima whereas DEUP-EI was able to reach the global maximum consistently. Experimental details are provided in Appendix H.3.

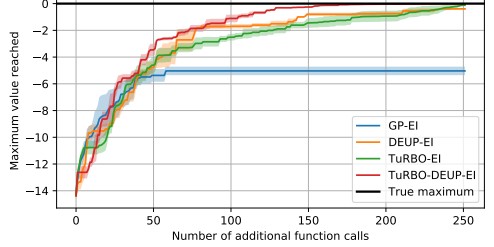
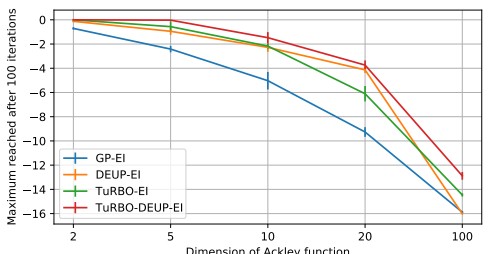

Figure 4: *Left.*Maximum value reached by the different optimization methods, for the 10 dimensional Ackley function. In each run, all the methods start with the same initial 20 points. The shaded areas represent the standard error across 3 runs. *Right.* maximum value reached in the budget-constrained setting, on the Ackley functions of different dimensions. The error bars represent the standard error across 3 different runs, with different initial sets of 20 pairs. The budget is 120 function calls in total. Higher is better and TuRBO-DEUP-EI is less hurt by dimensionality.

## 5.3 REINFORCEMENT LEARNING

Similar to sequential model optimization, a key challenge in reinforcement learning (RL) is efficient exploration of the input state space. To investigate the effectiveness of DEUP's uncertainty estimates in the context of RL, we incorporate epistemic uncertainties predicted by DEUP to DQN (Mnih et al., 2013), which we refer to as DEUP-DQN. Specifically, we train the uncertainty predictor with the objective to

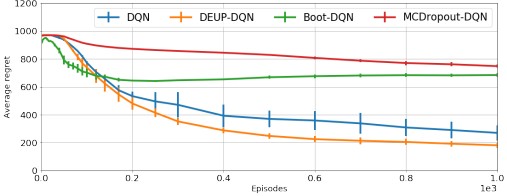

Figure 5: *Average regret on CartPole task. Error bars for standard error across 5 runs.*

predict the TD-error, using log-density as a stationarizing input. The predicted uncertainties are then used as an exploration bonus in the Q-values. Details of the experimental setup are in Appendix I. We evaluate DEUP-DQN on CartPole, a classic RL task from *bsuite* (Osband et al., 2020), against DQN + $\epsilon$-greedy, DQN + MC-Dropout (Gal & Ghahramani, 2016) and Bootstrapped DQN (Osband et al., 2016). Figure 5 shows that DEUP achieves lower regret faster, compared to all the baselines, which demonstrates the advantage of DEUP's uncertainty estimates for efficient exploration.

## 6 CONCLUSION AND FUTURE WORK

Whereas standard measures of epistemic uncertainty focus on variance, we argue that bias (misspecification) can also be reduced with predictors like neural nets, e.g. using early stopping. In a regression setup, the expected out-of-sample squared error minus aleatoric uncertainty thus captures all the uncertainty about $E[Y|x]$ that more data can allow to reduce. This motivates the DEUP estimator, using a second network trained to predict the errors of the first. In interactive settings, this nonetheless raises non-stationarity challenges for this estimator and we propose input features to reduce this issue and show experimentally their advantages. Future work should investigate ways to estimate aleatoric uncertainty when one cannot simply query the oracle several times on the same $x$.

ETHICS STATEMENT

The authors do not foresee any negative social impacts of this work, but of course the accumulation of improvements in ML could be misused as it may give more power to nefarious agents.

REPRODUCIBILITY STATEMENT

We provide the code to reproduce the results along with the submission. In addition to the code, we also outline all the details of the procedure to ensure reproducibility of the results. The general procedure for the method in the fixed-data and interactive setting is provided in Algorithm 2 and Algorithm 1 respectively. We also provide pseudocode for the specific instances of DEUP used for the Drug-Combinations and Reinforcement Learning experiments in Algorithm 4 and Algorithm 3. Detailed discussion about the methodology and hyperparameters are provided in the appendix in Appendix F, Appendix H, Appendix J and Appendix I. We provide proofs for all the propositions in Appendix C, with appropriate assumptions.

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

# APPENDICES

## A    RELATED WORK

Recently, several novel deep learning-based techniques to estimate uncertainty with a single model have been proposed. For example, Deep Evidential Regression Amini et al. (2020) is a method for estimating epistemic uncertainty that is based on a parametric estimate of model variance. This is in line with previous work using evidential uncertainty estimation (Malinin & Gales, 2018; Sensoy et al., 2018). Orthonormal Certificates, a set of learned features with a suitable loss function, are used in Tagasovska & Lopez-Paz (2019). These certificates capture the distance to the training set to learn an estimate of the epistemic uncertainty. This is further studied in Liu et al. (2020) who formalize *distance awareness*, which captures the model's ability to quantify the distance of a test sample from the training data manifold, as a necessary condition for uncertainty estimation. This distance awareness can be captured with a weight normalization step in training, in addition to using a GP as the output layer. DUE (van Amersfoort et al., 2021) is an instance of Deep Kernel Learning (Wilson et al., 2016), which is defined as a GP with a deep feature extractor inside the kernel. DUE improves upon SNGP by using an inducing point GP, and bi-lipschitz constraints on the feature extractor, giving better test set accuracies as well as improved training efficiency. DUN (Antoran et al., 2020) uses the disagreement between the outputs from intermediate layers as a measure of uncertainty. DUQ (van Amersfoort et al., 2020) on the other hand uses two-sided Jacobian regularization on RBF networks (LeCun et al., 1998) for reliable uncertainty estimates.

Wen et al. (2020) present an efficient way of implementing ensembles of neural networks, by using one shared matrix and a rank-1 matrix per member. The weights for each member are then computed as the Hadamard product of the shared matrix and the rank-1 matrix of the member. There has also been extensive work in scaling up Bayesian Neural Networks for high-dimensional data to capture epistemic uncertainty. SWAG (Maddox et al., 2019) fits a Gaussian distribution capturing the SWA (Izmailov et al., 2018) mean and a covariance matrix representing the first two moments of SGD iterated. This distribution is then used as a posterior over the neural network weights. Dusenberry et al. (2020) parametrize the BNN with a distribution on a rank-1 subspace for each weight matrix, inspired by BatchEnsembles (Wen et al., 2020). Vadera et al. (2020b); Malinin et al. (2020b) propose approaches to improve the efficiency of ensembles by distilling the distribution of predictions rather than the average, thus preserving the information about the uncertainty captured by the ensemble.

There is also a large body of work on methods for approximating samples from the Bayesian posterior on large datasets with efficient MCMC based approaches. (Welling & Teh, 2011; Zhang et al., 2020; Vadera et al., 2020a). The variance of this posterior distribution can then be computed and used as an uncertainty estimate. However, as we have discussed in the sections above, this does not account for model misspecification and thus is not an accurate estimate for the lack of knowledge of the predictor.

Kull & Flach (2015) present several decompositions of the total expected loss, including the decomposition into the epistemic and irreducible (aleatoric) loss. They present additive adjustments that reduce the scoring rules like the log-loss and Brier score, but do not tackle the general problem of uncertainty estimation.

There are also interesting connections between the problem of out-of-distribution generalization arising in sequential model optimization and Bayesian optimization, discussed here, and the possibility of reweighing examples, see Farquhar et al. (2021).

## B    EPISTEMIC UNCERTAINTY IN A GENERAL LOSS FUNCTION SETTING

We consider the setting presented in section 2, but with a general loss function $l$. We use the same notations as in section 2.

**Definition B.1** *The **total uncertainty** of $f$ at $x$ is defined as:*

$$\mathcal{U}(f, x) = \int l(f(x), y) dP(y|x).$$ (4)

**Definition B.2** *For a learning algorithm $\mathcal{L}$ which produces a distribution $P_{\mathcal{L}}(f(x)|\mathcal{D}_n)$ over possible solutions $f(x)$ at $x$, the* **model variance** *at $x$ is defined as*

$$V(\mathcal{L}, \mathcal{D}_n, x) = \int l(f(x), \hat{f}(x)) dP_{\mathcal{L}}(f(x)|\mathcal{D}_n)) \tag{5}$$

*with $\hat{f}(x) = \arg\min_{\bar{f}(x)} \int l(f(x), \bar{f}(x)) dP_{\mathcal{L}}(f(x)|\mathcal{D}_n))$. Note that for a loss function that is different from the square loss, the semantics of variance (such as its non-negativity) might be lost.*

Let us consider the special cases of the negative log-likelihood loss in general (for outputs which may be discrete or continuous) and that of the squared error loss (which ends up being a special case of the former for normally distributed outputs). Below we see $Q(Y|x)$ as a probability mass or density function (over $y$), which is also a function of $x$.

**Definition B.3** *The negative log-likelihood (NLL) loss takes as first argument $Q(Y|x)$ a probability mass or density function and returns*

$$l_{NLL}(Q(Y|x), y) = -\log Q(Y = y|x). \tag{6}$$

**Proposition 2** *For the NLL loss with ground truth $P(Y|x)$ and predictor $Q(Y|x)$, the total uncertainty $\mathcal{U}(Q(Y|x), x)$ is a cross-entropy, i.e.,*

$$\mathcal{U}(Q(Y|\,.\,), x) = CE(P(Y|x)||Q(Y|x))$$
$$= -\int dP(y|x) \log Q(y|x) \tag{7}$$

The proposition is shown by applying the definitions.

**Proposition 3** *For the NLL loss with ground truth $P(Y|x)$, the aleatoric uncertainty $\mathcal{A}(x)$ in this setting is the entropy $H[P(Y|x)]$ of the ground truth conditional:*

$$\mathcal{A}(x) = -\int dP(y|x) \log P(y|x) = H[P(Y|x)], \tag{8}$$

The proposition is shown from the cross-entropy $CE(P(Y|x)||Q(Y|x))$ being minimized when $Q = P$.

**Proposition 4** *For the NLL loss with ground truth $P(Y|x)$ and predictor $Q(Y|x)$, the epistemic uncertainty $\mathcal{E}(Q(Y|x), x)$ is the Kullback-Liebler divergence between $P$ and $Q$ (with $P$ as the reference):*

$$\mathcal{E}(Q(Y|\,.\,), x) = KL(P(Y|x)||Q(Y|x))$$
$$= \int dP(y|x) \log \frac{P(y|x)}{Q(y|x)} \tag{9}$$

The proposition is shown by combining the above two propositions and the definition of epistemic uncertainty.

To move towards the MSE loss, consider the special case of NLL with a conditionally Normal output density for both $P$ and $Q$.

**Proposition 5** *For the NLL loss with a conditionally Normal output density for both $P$ and $Q$, with respective means $f^*(x)$ and $\hat{f}(x)$ and respective variances $\sigma_P^2(x)$ and $\sigma_Q^2(x)$, the epistemic uncertainty is*

$$\mathcal{E}(Q(Y|\,.\,), x) = \frac{1}{2\sigma_Q^2(x)} l_{MSE}(\hat{f}(x), f^*(x))$$
$$+ KL(P(Y|x)||\tilde{Q}(Y|x)), \tag{10}$$

*where $\tilde{Q}(\,.\,|x)$ is obtained by shifting $Q(\,.\,|x)$ towards $P(\,.\,|x)$ (i.e., $\tilde{Q}(\,.\,|x)$ is Gaussian with mean $f^*(x)$ and variance $\sigma_Q^2(x)$ ), and the Bayes-optimal mean predictor is $f^*(x) = E_P[Y|x]$. Note that if $\sigma_P = \sigma_Q$, then the KL term is zero.*

The proof is presented in Appendix C. We can compare with the MSE loss (which assumes a constant variance $\sigma = \sigma_P = \sigma_Q$) and obtain the same result up to variance scaling.

## C    PROOFS

### C.1    PROPOSITION 1

It is a well known result that, because $f^*(x)$ is the mean of $P(.|x)$, it is also the minimizer of $\hat{y} \mapsto \int (\hat{y} - y)^2 dP(y|x)$. $f^*$ is thus a Bayes Optimal predictor.

By definition of the total uncertainty:

$$\mathcal{U}(f,x) = \int (f(x) - y)^2 dP(y|x) = E[(f(x) - Y)^2].$$

Hence, by definition of aleatoric uncertainty:

$$\mathcal{A}(x) = \mathcal{U}(f^*, x) = E[(f^*(x) - Y)^2].$$

and by definition of epistemic uncertainty

$$
\begin{aligned}
\mathcal{E}(f,x) &= E\left[(f(x) - y)^2 - (f^*(x) - y)^2\right] \\
&= f(x)^2 - f^*(x)^2 - 2(f(x) - f^*(x))f^*(x) \\
&= (f(x) - f^*(x))^2.
\end{aligned}
$$

Which concludes the proof.

### C.2    PROPOSITION 5

From Equation 9, we get:

$$
\begin{aligned}
\mathcal{E}(Q(Y|\,.\,),x) &= KL(P(Y|x)||Q(Y|x)) \\
&= \log \frac{\sigma_Q(x)}{\sigma_P(x)} + \frac{\sigma_P^2(x) + (f(x) - f^*(x))^2}{2\sigma_Q^2(x)} - \frac{1}{2} \\
&= \frac{1}{2\sigma_Q^2(x)} l_{MSE}(f(x), f^*(x)) + \log \frac{\sigma_Q(x)}{\sigma_P(x)} + \frac{\sigma_P^2(x) + (f^*(x) - f^*(x))^2}{2\sigma_Q^2(x)} - \frac{1}{2} \\
&= \frac{1}{2\sigma_Q^2(x)} l_{MSE}(f(x), f^*(x)) + KL(P(Y|x)|\tilde{Q}(Y|x))
\end{aligned}
$$

Which concludes the proof

## D    ESTIMATING ALEATORIC UNCERTAINTY WITH ACCESS TO AN ORACLE

In scenarios like active learning, one has access to an oracle from which we can obtain samples of $Y \sim P(Y|x)$ at any given point $x$. In that case, one can train an estimator $a(x)$ of aleatoric uncertainty by obtaining $K > 1$ samples $y_1, \ldots y_K$ at the same $x$, for a set of representative $x$'s.

More formally, if we have multiple independent outcomes $y_1, \ldots, y_K \sim P(Y|x)$ for each input point $x$, then training a predictor $a$ with the squared loss on (input, target) examples $\left(x, \frac{K}{K-1}\overline{Var}(y_1, \ldots, y_K)\right)$, where $\overline{Var}$ denotes the empirical variance, yields an estimator of the aleatoric uncertainty.

Naturally, this estimator is asymptotically unbiased, if the learning algorithm ensures asymptotic convergence to a Bayes-Optimal predictor.

This is due to the fact that $E_{y_1, \ldots, y_K}\left[\frac{K}{K-1}\overline{Var}(y_1, \ldots, y_K)\right] = Var_{P(y|x)}[y|x]$, which according to Proposition 1 is equal to $\mathcal{A}(x)$.

# E  PSEUDO CODES

Algorithm 2 illustrates the training procedure when a held-out validation set is available. We focus on $y \in \mathbb{R}$ in this paper because it makes sense for active learning applied to black-box optimization (where we want to maximize it) but the algorithms can trivially be applied to the case where $y \in \mathbb{R}^d$ for any $d$. Similarly, the algorithms can be generalized to other losses besides the square loss, following the generalized theory presented above in Appendix B.

---

**Algorithm 2** DEUP with a fixed training set: Training procedure to obtain estimates of epistemic uncertainty

---

**Data:** $\mathcal{D}$ the training dataset with pairs $(x, y)$ with $x \in \mathcal{X}$, $y \in \mathbb{R}$; $\mathcal{D}_{out}$ the out-of-sample dataset with pairs $(x, y)$, to train the uncertainty estimator
$\mathcal{X}$, the input/search space
$a : \mathcal{X} \mapsto \mathbb{R}$, trained estimator of aleatoric uncertainty (
$f : \mathcal{X} \mapsto \mathbb{R}$, main predictor, trained on $\mathcal{D}$
$e : \mathcal{X} \mapsto \mathbb{R}$, total uncertainty estimator (estimates error of $f$)
**Training:**
Initialize empty dataset of errors $\mathcal{D}_e$
**Cross-validation step (Optional, if using other features for the error predictor):** Similar to the corresponding step of Algorithm 1.
**for** *every pair $(x, y)$ in $\mathcal{D} \cup \mathcal{D}_{out}$* **do**
$\quad | \quad \mathcal{D}_e \leftarrow \mathcal{D}_e \cup \{(x, (y - f(x))^2)\}$
**end**
Fit $e$ on $\mathcal{D}_e$
**Evaluation:** For every input $x$, return $e(x) - a(x)$ as an estimator of epistemic uncertainty at $x$

---

# F  REJECTING DIFFICULT EXAMPLES

We adapt the standard OOD rejection task (van Amersfoort et al., 2020; Liu et al., 2020) to measure the Spearman Rank Correlation of the predicted uncertainty with the true generalization error, in addition to the OOD Detection AUROC. We use MC-Dropout (Gal & Ghahramani, 2016), Deep Ensemble (Lakshminarayanan et al., 2017), DUE(van Amersfoort et al., 2021) and DUQ van Amersfoort et al. (2020) as the baselines [2]. We use these baselines as representatives for the major approaches for uncertainty estimation in recent literature. For all the methods, including DEUP we consider two architectures for the main predictor, ResNet-18 and ResNet-50 (He et al., 2016) (Table 2) to study the effect of model capacity. Note that for the ResNet50 DEUP model we continue using the ResNet-18 based DUE as variance source.

Table 2: Spearman Rank Correlation between predicted uncertainty and the true generalization error on OOD data (SVHN) with ResNet-50 models (3 seeds) trained on CIFAR-10.

| Model | ResNet-50 |
|---|---|
| MC-Dropout | $0.312 \pm 0.003$ |
| Deep Ensemble | $0.401 \pm 0.004$ |
| DUQ | $0.399 \pm 0.003$ |
| **DEUP (D+V)** | $\mathbf{0.465 \pm 0.002}$ |

**Training**  The baselines were trained with the CIFAR-10 training set with $10\%$ set aside as a validation set for hyperparameter tuning. The hyperparameters are presented in Table 3 and Table 4. The hyperparameters not specified are set to the default values. For DEUP, we consider the log-density, model-variance estimate and the seen-unseen bit as the features for the error predictor. The density

---

[2]MC-Dropout and Deep Ensemble baselines are based on https://github.com/google/uncertainty-baselines, DUQ based on https://github.com/y0ast/deterministic-uncertainty-quantification and DUE based on https://github.com/y0ast/DUE

estimator we use is Masked-Autoregressive Flows (Papamakarios et al., 2017) and the variance estimator used is DUE (van Amersfoort et al., 2021). Note that as indicated earlier $x$, the input image, is not used as a feature for the error predictor. We present those ablations in the next sub-section. For training DEUP, the CIFAR-10 training set is divided into 5 folds, with each fold containing 8 unique classes. For each fold, we train an instance of the main predictor, density estimator and model variance estimator on only the corresponding 8 classes. The remaining 2 classes act as the out-of-distribution examples for training the error predictor. Using these folds we construct a dataset for training the error predictor, a simple feed forward network. The error predictor is trained with the $\log$ targets (i.e. log MSE between predicted and observed error). This helps since the scale of the errors varies over multiple orders of magnitude. We then train the main predictor, density estimator and the variance estimator on the entire CIFAR-10 dataset, for evaluation. The hyperparameters are presented in Table 4. For all models, we train the main predictor for 75 and 125 epochs for ResNet-18 and ResNet-50 respectively. We use SGD with Momentum (set to 0.9), with a multi-step learning schedule with a decay of 0.2 at epochs $[25, 50]$ and $[45, 90]$ for ResNet-18 and ResNet-50 respectively. One complete training run for DEUP takes about 1.5-2 GPU days on a V100 GPU. In total these set of experiments took about 31 GPU days on a Nvidia V100 GPU.

Table 3: **Left**: Hyperparameters for training Deep Ensemble (Lakshminarayanan et al., 2017). **Right**: Hyperparameters for training MC-Dropout (Gal & Ghahramani, 2016).

| Parameters | Model | |
|---|---|---|
| | ResNet-18 | ResNet-50 |
| Number of members | 5 | 5 |
| Learning Rate | 0.05 | 0.01 |

| Parameters | Model | |
|---|---|---|
| | ResNet-18 | ResNet-50 |
| Number of samples | 50 | 50 |
| Dropout Rate | 0.15 | 0.1 |
| L2 Regularization Coefficient | 6e-5 | 8e-4 |
| Learning Rate | 0.05 | 0.01 |

Table 4: **Left**: Hyperparameters for training DUQ (van Amersfoort et al., 2020). **Right**: Hyperparameters for training DUE (van Amersfoort et al., 2021).

| Parameters | Model | |
|---|---|---|
| | ResNet-18 | ResNet-50 |
| Gradient Penalty | 0.5 | 0.65 |
| Centroid Size | 512 | 512 |
| Length scale | 0.1 | 0.2 |
| Learning Rate | 0.05 | 0.025 |

| Parameters | Model |
|---|---|
| | ResNet-18 |
| Inducing Points | 50 |
| Kernel | RBF |
| Lipschitz Coefficient | 2 |
| BatchNorm Momentum | 0.99 |
| Learning Rate | 0.05 |
| Weight Decay | 0.0005 |

**Ablations** We also perform some ablation experiments to study the effect of each feature for the error predictor. The Spearman rank correlation coefficient between the generalization error and the variance feature, $V$, from DUE van Amersfoort et al. (2021) alone is $37.84 \pm 0.04$, and the log-density, $D$, from MAF Papamakarios et al. (2017) alone is $30.52 \pm 0.03$. With only the image ($x$) the SRCC is $36.58 \pm 0.16$

Table 6 presents the results for these experiments. We observe that combining all the features performs the best. Also note that using the log-density and variance as features to the error predictor we observe better performance than using them directly, indicating that the error predictor perhaps captures

Table 5: Hyperparameters for training DEUP.

| Parameters | Model | |
|---|---|---|
| | ResNet-18 | ResNet-50 |
| Uncertainty Predictor Architecture | [1024] x 5 | [1024] x 5 |
| Uncertainty Predictor Epochs | 100 | 100 |
| Uncertainty Predictor LR | 0.01 | 0.01 |
| Main Predictor Learning Rate | 0.05 | 0.01 |

a better target for the epistemic uncertainty. The boolean feature ($B$) indicating seen examples, discussed in Section 3.2, also leads to noticeable improvments.

Table 6: Spearman Rank Correlation between predicted uncertainty and the true generalization error on OOD data (SVHN) with variants of DEUP with different features as input for the uncertainty predictor. $D$ indicates the log-density from MAF Papamakarios et al. (2017), $V$ indicates variance from DUQ van Amersfoort et al. (2020) and $B$ indicates a bit indicating if the data is seen.

| Features | Model | |
|---|---|---|
| | ResNet-18 | ResNet-50 |
| $D+V+B$ | $\mathbf{0.426 \pm 0.009}$ | $\mathbf{0.465 \pm 0.002}$ |
| $D+V$ | $0.419 \pm 0.003$ | $0.447 \pm 0.003$ |
| $V+B$ | $0.401 \pm 0.004$ | $0.419 \pm 0.004$ |
| $D+B$ | $0.403 \pm 0.003$ | $0.421 \pm 0.002$ |

### F.1 PREDICTING UNCERTAINTY UNDER DISTRIBUTION SHIFT

We also consider the task of uncertainty estimation in the setting of shifted distributions (Ovadia et al., 2019; Hendrycks & Dietterich, 2019). We evaluate the uncertainty predictions of models trained with CIFAR-10, on CIFAR-10-C (Hendrycks & Dietterich, 2019) which consists of images from CIFAR-10 distorted using 16 corruptions like gassian blur, impulse noise, among others. Figure 6 shows that even in the shifted distribution setting, the uncertainty estimates of DEUP correlate much better with the error made by the predictor, compared to the baselines.

## G  DEUP IN THE PRESENCE OF ALEATORIC UNCERTAINTY

In the presence of aleatoric uncertainty, we have seen that DEUP's error predictor $e$ is an estimate of the total uncertainty, rather than the epistemic uncertainty. However, if we have access to to an estimator of aleatoric uncertainty $a$, then $e - a$ becomes an estimator of epistemic uncertainty. To show the difference in behavior of DEUP when there is aleatoric uncertainty, we consider, a modified version of the experiment in Section 5.1.1, with a non-deterministic oracle (ground truth function). Because of the noisy training dataset, GP conflates epistemic and aleatoric uncertainty, which makes the gap between the predicted epistemic uncertainty (as measured by the GP variance) and the true epistemic uncertainty (as measured by the MSE between the GP mean and the noiseless ground truth function) higher than in the deterministic setting of Section 5.1.1.

Similarly, in order to train DEUP's uncertainty estimator, more out-of-sample data is needed compared to the noiseless setting. In figure 7, we consider the setting described in Section D, and train a separate predictor on the targets $y_1, \ldots, y_k$ to estimate the aleatoric uncertainty (which boils down to the variance of the ground-truth function). Because these targets are themselves noisy, we chose a simple linear regressor as the estimator of aleatoric uncertainty, to avoid overfitting to the noise.

A key distinction of this setting, is that DEUP's training data (the errors of the main predictor) are themselves noisy, which makes it important to use more out-of-sample data to obtain reasonable total uncertainty estimates (from which we subtract the estimates of the aleatoric uncertainty).

## H  SEQUENTIAL MODEL OPTIMIZATION EXPERIMENTS

We use BoTorch[3] (Balandat et al., 2020) as the base framework for our experiments.

For all our Sequential Optimization algorithms, we use Algorithm 1 to train DEUP uncertainty estimators. We found that the optional step of pre-filling the uncertainty estimator dataset $\mathcal{D}_e$ was important given the low number of available training points. We used half the initial training set (randomly chosen) as in-sample examples (used to train the main predictor and an extra-feature generator) and the other half as out-of-sample examples to provide instances of high epistemic

---

[3]https://botorch.org/

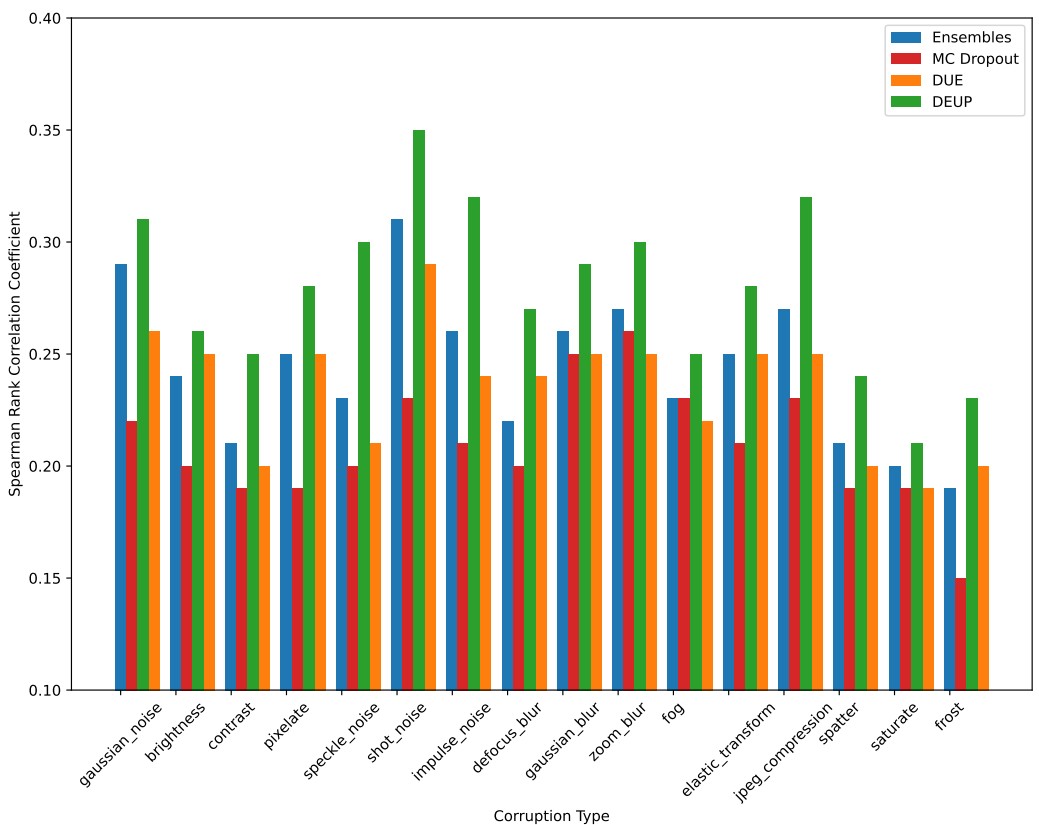

Figure 6: Spearman Rank Correlation Coefficient between the predicted uncertainty and true error for models trained with CIFAR-10, and evaluated on CIFAR-10-C. DEUP outperforms the baselines on all types of corruptions.

uncertainty to train an uncertainty predictor; we repeated the procedure by alternating the roles of the two halves of the dataset. We repeated the whole procedure twice using a new random split of the dataset, thus ending up with 4 training points in $\mathcal{D}_e$ for every initial training point in $\mathcal{D}_{init}$.

The error predictor is trained with the log targets (i.e. log MSE between predicted and observed error). This helps since the scale of the errors varies over multiple orders of magnitude.

Computationally, the training time of DEUP-EI depends on various choices (e.g. the features used to train the epistemic uncertainty predictor, the dimension of the input, the learning algorithms, etc..). Additionally, the training time for the uncertainty predictor varies at each step of the optimization. In total, the sequential optimization experiments took about 1 CPU day.

### H.1 ONE-DIMENSIONAL FUNCTION TOY EXAMPLE

In Figure 8, we show the results of DEUP-EI, compared to GP-EI, MCDropout-EI and Ensembles-EI, in the task of optimizing a synthetic one-dimensional function. Because MCDropout and Ensembles are trained on in-sample data only, they are unable to generalize their uncertainty estimates, which makes them bad candidates for Sequential Model Optimization, because they are easily stuck in local minima, and require many iterations before the acquisition function gives more weight to the predicted uncertainties than the current maximum.

For Random acquisition, we sampled for different seeds 56 points, and used the (average across the seeds of the) maximum of the first 6 values as the first value in the plots (Figures 4 and 8). Note that

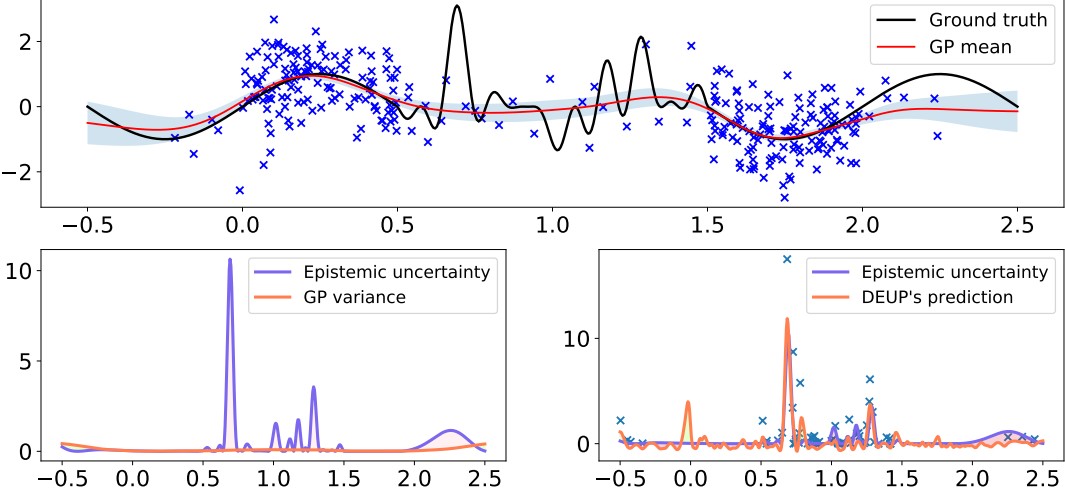

Figure 7: *Top.* A GP is trained to regress a function using noisy samples. GP uncertainty (model standard deviation) is shaded in blue. *Bottom left.* Using GP variance as a proxy for epistemic uncertainty misses out on more regions of the input space, when compared to Figure 1 . *Bottom right.* Using additional out-of-sample data in low density regions, a second GP is trained to predict the generalization error of the first GP (total uncertainty). Using second samples from the oracle for each of the training points, a linear regressor fits the training pairs $(x, \frac{1}{2}(y_1 - y_2)^2)$ to estimate the pointwise aleatoric uncertainty (constant in this case). The aleatoric uncertainty is subtracted from DEUP's (second GP) predictions to obtain more accurate of epistemic uncertainty. Note that no constraint is imposed on DEUP's outputs, which explains the predicted negative values for uncertainties. In practice, if these predicted uncertainties were to be used, (soft) clipping should be used.

because the function is specifically designed to have multiple local maxima, GP-EI also required more optimization steps, and actually performed worse than random acquistion

As a stationarizing input feature, we used the variance of a GP fit on the available data at every step. We found that the binary (in-sample/out-of-sample) feature and density estimates were redundant with the variance feature and didn't improve the performance as captured by the number of additional function calls. We used a GP for the DEUP uncertainty estimator. Using a neural net provided similar results, but was computationally more expensive in this 1-D case with few datapoints. We used a 3-hidden layer neural network, with 128 neurons per layer and a ReLU activation function, with Adam (Kingma & Ba, 2015) and a learning rate of $10^{-3}$ (and default values for the other hyperparameters) to train the main predictor for DEUP-EI (in order to fit the available data). The same network architecture and learning rate were used for the Dropout and Ensemble baselines. We used 3 networks for the Ensemble baseline, and a dropout probability of 0.3 for the Dropout baseline, with 100 test-time forward passes to compute uncertainty estimates.

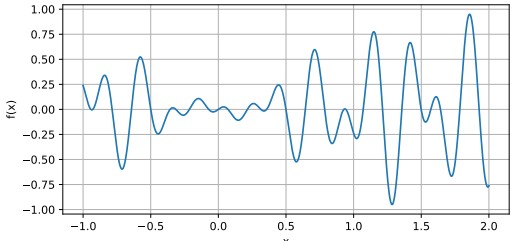 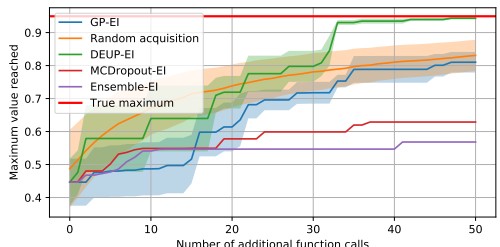

Figure 8: *Left.* Synthetic function to optimize. *Right.* Maximum value reached by the different methods on the synthetic function. The shaded areas represent the standard error across 5 different runs, with different initial sets of 6 pairs. For clarity, the shaded areas are omitted for the two worst performing methods. In each run, all the methods start with the same initial set of 6 points. GP-EI tends to get stuck in local optima and requires more than 50 steps, on average, to reach the global maximum.

## H.2    TWO-DIMENSIONAL FUNCTION

To showcase DEUP's usefulness for Sequential Model Optimization in with a number of dimensions greater than 1, we consider the optimization of the Levi N.13 function, a known benchmark for optimization. The function $f$ takes a point $(x, y)$ in 2D space and returns:

$$f(x, y) = -\left(\sin^2(3\pi x) + (x-1)^2(1 + \sin^2(3\pi y)) + (y-1)^2(1 + \sin^2(2\pi y))\right)$$

We use the box $[-10, 10]^2$ as the optimization domain. In this domain, the maximum of the function is 0, and it is reached at $(1, 1)$. The function has multiple local maxima, as shown in Figure 9(a)[4].

Similar to the previous one-dimensional function, MCDropout and Ensemble provided bad performances and are omitted from the plot in 9(b). We used the same setting and hyperparameters for DEUP as for the previous function. DEUP-EI is again the only method that reaches the global maximum consistently in under 56 function evaluations.

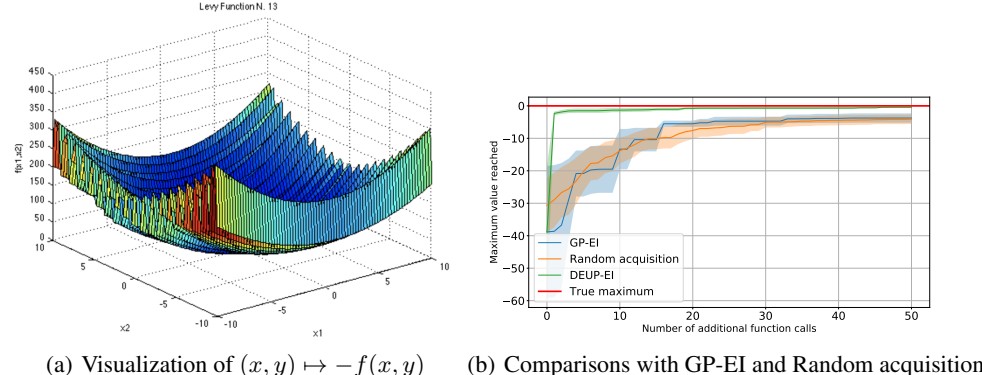

(a) Visualization of $(x, y) \mapsto -f(x, y)$    (b) Comparisons with GP-EI and Random acquisition

Figure 9: Sequential Model Optimization on the Levi N.13 function

---

[4]Plot of the function copied from https://www.sfu.ca/ ssurjano/levy13.html

### H.3 ADDITIONAL DETAILS FOR THE ACKLEY FUNCTION EXPERIMENT, FOR SYNTHETIC DATA IN HIGHER DIMENSIONS

The Ackley function of dimension $d$ is defined as:

$$Ackley_d : \mathcal{B} \to \mathbb{R}$$

$$x \mapsto A \exp\left(-B\sqrt{\frac{1}{d}\sum_{i=1}^{d}x_i^2}\right) + \exp\left(\frac{1}{d}\sum_{i=1}^{d}cos(cx_i)\right) - A - \exp(1)$$

where $\mathcal{B}$ is a hyperrectangle of $\mathbb{R}^d$. $(0,\dots,0)$ is the only global optimizer of $Ackley_d$, at which the function is equal to 0. We use BoTorch's default values for $A, B, c$, which are $20, 0.2, 2\pi$ respectively.

In our experiments, we used $\mathcal{B} = [-10, 15]^d$ for all dimensions $d$.

For the TurBO baseline, we use BoTorch's default implementation, with Expected Improvement as an acquisition function, and a batch size of 1 (i.e. acquiring one point per step).

For fair comparisons, for DEUP, we use a Gaussian Process as the main model, and use its variance as the only input of the epistemic uncertainty predictor. This means that we calibrate the GP variance to match the out-of-sample squared error, using another GP to perform the regression. TurBO-DEUP is a combination of both, in which we perform the variance calibration task for the local GP models of TurBO. The uncertainty predictor, i.e. the GP regressor, is trained with $\log$ targets, as in Appendix H.1, but also with $\log$ variances as inputs.

Note that only the stationarizing feature is used as input for the uncertainty predictor. When we used the input $x$ as well, we found that the GP error predictor overfits on the $x$ part of the input $(x, v)$, and it was detrimental to the final performances. For all experiments, we used 20 initial points.

## I REINFORCEMENT LEARNING EXPERIMENTS

For RL experiments, we used *bsuite* (Osband et al., 2020), a collection of carefully designed RL environments. *bsuite* also comes with a list of metrics which aim to evaluate RL agents from different aspects. We compare the agents based on the *basic* metric and average regret as they capture both sample complexity and final performance. The default DQN agent is used as the base of our experiments with a 3 layer fully-connected (FC) neural network as its Q-network. For the Bootstrapped DQN baseline, we used the default implementation provided by *bsuite*. To implement DQN + MC-Dropout, following the implementation from Gal & Ghahramani (2016), two dropout layers with dropout probability of 0.1 are used before the second and the third FC layers. In order take an action, the agent performs a single stochastic forward pass through the Q-network, which is equivalent to taking a sample from the posterior over the Q-values, as done in Thompson sampling, an alternative to $\epsilon-$greedy exploration.

As a density estimator, we used a Kernel Density Estimator (KDE) with a Gaussian kernel and bandwidth of 1 to map states to densities. This KDE is fit after each 10000 steps (actions) with a batch of samples from the replay buffer (which is of size 10000). The uncertainty estimator network (E-network) has the same number of layers as the Q-network, with an additional Softplus layer at the end. All other hyperparameters are the same as the default implementation by Osband et al. (2020). One complete training run for the DEUP-DQN with 5 seeds experiments takes about 0.04-0.05 GPU days on a V100 GPU. In total RL experiments took about 0.15 GPU days on a Nvidia V100 GPU.

---

**Algorithm 3** DEUP-DQN

---

Initialize replay buffer $\mathcal{D}$ with capacity $\mathcal{N}$
$Q_\theta(s, a)$: state-action value predictor
$E_\phi(\log d)$: uncertainty estimator network, which takes the log-density of the states as the input
$d(s)$: Kernel density estimator (KDE)
K: KDE fitting frequency
W: Number of warm-up episodes
**for** *episode=1 to M* **do**
    set $s_0$ as the initial state
    **for** *t=1 to max-steps-per-episode* **do**
        **with probability** $\epsilon$**:** take a random action, **otherwise:**
        **if** $episode \leq$ W: $a = max_a Q_\theta(s_t, a)$, **else:** $a = max_a \left[ Q_\theta(s_t, a) + \kappa \times E_\phi(\log d(s_t))(a) \right]$
        store $(s_t, a_t, r_t, s_{t+1})$ in $\mathcal{D}$
        Sample random minibatch B of transitions $(s_j, a_j, r_j, s_{j+1})$ from $\mathcal{D}$
        **if** $s_j$ is a final state: $y_j = r_j$, **else:** $y_j = r_j + \gamma max_a Q(s_t, a)$
        **Update Q-network:**
$$\theta \leftarrow \theta + \alpha_Q . \nabla_\theta \, \mathbb{E}_{(s,a) \sim B} \left[ \left( y_j - Q_\theta(s, a) \right)^2 \right]$$
        **Update E-network:**
$$\phi \leftarrow \phi + \alpha_E . \nabla_\phi \, \mathbb{E}_{(s,a) \sim B} \left[ \left[ \left( y_j - Q_\theta(s, a) \right)^2 - E_\phi(\log d(s_t))(a) \right]^2 \right]$$
        **if** mod(*total-steps*, K) = 0: fit the KDE $d$ on the states of $\mathcal{D}$
    **end**
**end**

---

## J DRUG COMBINATION EXPERIMENTS

To validate DEUP's uncertainty estimates in a real-world setting, we measured its performance on a regression task predicting the synergy of drug combinations. While much effort in drug discovery is spent on finding novel small molecules, a potentially cheaper method is identifying combinations of pre-existing drugs which are synergistic (i.e., work well together). Indeed, drug combinations are the current standard-of-care for a number of diseases including HIV, tuberculosis, and some cancers Cihlar & Fordyce (2016); Organization & Initiative (2010); Mokhtari et al. (2017).

However, due to the combinatorial nature of drug combinations, identifying pairs exhibiting synergism is challenging. Compounding this problem is the high monetary cost of running experiments on promising drug combinations, as well as the length of time the experiments take to complete. Uncertainty models could be used by practitioners downstream to help accelerate drug combination treatment discoveries and reduce involved development costs.

To test DEUP's performance on this task we used the DrugComb and LINCS L1000 datasets Zagidullin et al. (2019); Subramanian et al. (2017). DrugComb is a dataset consisting of pairwise combinations of anti-cancer compounds tested on various cancer cell lines. For each combination, the dataset provides access to several synergy scores, each indicating whether the two drugs have a synergistic or antagonistic effect on cancerous cell death. LINCS L1000 contains differential gene expression profiles for various cell lines and drugs. Differential gene expressions measure the difference in the amount of mRNA related to a set of influential genes before and after the application of a drug. Because of this, gene expressions are a powerful indicator of the effect of a single drug at the cellular level.

In our experiments, each drug is represented by its Morgan fingerprint Morgan (1965)[5] (with 1,024 bits and a radius of 3) as well as two differential gene expression profiles (each of dimension 978) from two cell lines (PC-3 and MCF-7). In order to use gene expression features for every drug, we only used drug pairs in DrugComb where both drugs had differential gene expression data for cell lines PC-3 and MCF-7.

---

[5]The Morgan fingerprint represents a molecule by associating with it a boolean vector specifying its chemical structure. Morgan fingerprints have been used as a signal of various molecular characteristics to great success Ballester & Mitchell (2010); Zhang et al. (2006).

We first compared the quality of DEUP's uncertainty estimations to other uncertainty estimation methods on the task of predicting the combination sensitivity score Malyutina et al. (2019) for drug pairs tested on the cell line PC-3 (1,385 examples). We evaluated the uncertainty methods using a train, validation, test split of 40%, 30%, and 30%, respectively. The underlying model used by each uncertainty estimation method consisted of a *single drug* fully connected neural network (2 layers with 2048 hidden units and output of dimension 1024) and a *combined drug* fully connected neural network (2 layers, with 128 hidden units). The embeddings of an input drug pair's drugs produced by the *single drug* network are summed and passed to the *combined drug* network, which then predicts final synergy. By summing the embeddings produced by the *single drug* network, we ensure that the model is invariant to permutations in order of the two drugs in the pair. The models were trained with Adam Kingma & Ba (2015), using a learning rate of 1e-4 and weight decay of 1e-5. For MC-Dropout we used a dropout probability of $0.1$ on the two layers of the *combined drug* network and 3 test-time forward passes to compute uncertainty estimates. The ensemble used 3 constituent models for its uncertainty estimates. Both Ensemble and MC-Dropout models were trained with the *MSE* loss.

We also compared against DUE (van Amersfoort et al., 2021) which combines a neural network feature extractor with an approximate Gaussian process. Spectral normalization was added to all the layers of the *combined drug* network and of the *single drug* network. Let $d_{\text{emb}}$ denote the dimension of the output of the *combined drug* network, which is also the input dimension of the approximate Gaussian process. We conducted a grid-search over different values of $d_{\text{emb}}$ (from 2 to 100), the number of *inducing points* (from 3 to 200), the learning rate, and the kernel used by the Gaussian process. The highest correlation of uncertainty estimates with residuals was attained with $d_{\text{emb}} = 10$, 100 *inducing points*, a learning rate of 1e-2, and the *Matern12* kernel.

---

**Algorithm 4** DEUP for Drug Combinations

---

**Data:** $\mathcal{D}$ dataset of pairwise drug combinations, along with synergy scores $((d_1, d_2), y)$
**Initialization:**
Split training set into two halves, *in-sample* $\mathcal{D}_{in}$ and *out-of-sample* $\mathcal{D}_{out}$
$f_\mu(d_1, d_2)$: $\hat{\mu}$ predictor which takes a pair of drugs as input
$f_\sigma^{in}(d_1, d_2)$: In-sample $\hat{\sigma}_{in}$ error predictor
$f_\sigma^{out}(d_1, d_2)$: Out-of-sample $\hat{\sigma}_{out}$ error predictor
**Training:**
**while** *training not finished* **do**
  *In-sample update*
  Get an *in-sample* batch $(d_{1,in}, d_{2,in}, y_{in}) \sim \mathcal{D}_{in}$
  Predict $\hat{\mu} = f_\mu(d_{1,in}, d_{2,in})$ and *in-sample* error $\hat{\sigma}_{in} = f_\sigma^{in}(d_{1,in}, d_{2,in})$
  Compute *NLL*: $\frac{\log(\hat{\sigma}_{in}^2)}{2} + \frac{(\hat{\mu}-y_{in})^2}{2\hat{\sigma}_{in}^2}$
  Backpropagate through $f_\mu$ and $f_\sigma^{in}$ and update.
  *Out-of-sample update*
  Get an *out-of-sample* batch $(d_{1,out}, d_{2,out}, y_{out}) \sim \mathcal{D}_{out}$
  Estimate $\hat{\mu} = f_\mu(d_{1,out}, d_{2,out})$ and *out-of-sample* error $\hat{\sigma}_{out} = f_\sigma^{out}(d_{1,out}, d_{2,out})$
  Compute *NLL*: $\frac{\log(\hat{\sigma}_{out}^2)}{2} + \frac{(\hat{\mu}-y_{out})^2}{2\hat{\sigma}_{out}^2}$
  Backpropagate through $f_\sigma^{out}$ and update.
**end**

---

The DEUP model we used outputs two heads $\begin{bmatrix} \hat{\mu} \\ \hat{\sigma} \end{bmatrix}$ and is trained with the *NLL* $\frac{\log(\hat{\sigma}^2)}{2} + \frac{(\hat{\mu}-y)^2}{2\hat{\sigma}^2}$ in a similar fashion as in Lakshminarayanan et al. (2017). To obtain a predictor of the out-of-sample error, we altered our optimization procedure so that the $\mu$ and $\sigma$ heads were not backpropagated through at all times. Specifically, we first split the training set into two halves, terming the former the in-sample set $\mathcal{D}_{in}$ and the latter the out-of-sample set $\mathcal{D}_{out}$. We denote as $f_\sigma^{in}$ the in-sample error predictor and $f_\sigma^{out}$ the out-of-sample error predictor. $f_\sigma^{out}$ is used to estimate total uncertainty. Note that in this setting, $f_\sigma^{out}$ predicts the square root of the epistemic uncertainty ($\hat{\sigma}_{out}$) rather than the epistemic uncertainty itself ($\hat{\sigma}_{out}^2$).

In our experiments, an extra bit is added as input to the model in order to indicate whether a given batch is from $\mathcal{D}_{in}$ or $\mathcal{D}_{out}$. Through this, the same model is used to estimate $f_\sigma^{in}$ and $f_\sigma^{out}$ with the model estimating $f_\sigma^{in}$ when the bit indicates an example is drawn from $\mathcal{D}_{in}$ and $f_\sigma^{out}$ otherwise. When

the batch is drawn from $\mathcal{D}_{in}$, both heads are trained using NLL using a single forward pass. However, when the data is drawn from $\mathcal{D}_{out}$ only the $\hat{\sigma}$ head is trained. To do this, we must still predict $\hat{\mu}$ in order to compute the NLL. But the $\hat{\mu}$ predictor $f_\mu$ must be agnostic to the difference between $\mathcal{D}_{in}$ and $\mathcal{D}_{out}$. To solve this, we perform two separate forward passes. The first pass computes $\hat{\mu}$ and sets the indicator bit to 0 so $f_\mu$ has no notion of $\mathcal{D}_{out}$, while the second pass computes $\hat{\sigma}$, setting the bit to 1 to indicate the true source of the batch. Finally, we backpropagate through the $\hat{\sigma}$ head only. The training procedure is described in Algorithm 4

We report several measures for the quality of uncertainty predictions on a separate test set in Table 7.

| Model | Corr. w. res. | U. Bound | Ratio | Log Likelihood | Coverage Probability | CI width |
|---|---|---|---|---|---|---|
| MC-Dropout | $0.14 \pm 0.07$ | $0.56 \pm 0.05$ | $0.25 \pm 0.12$ | $-20.1 \pm 6.8$ | $11.4 \pm 0.2$ | $3.1 \pm 0.1$ |
| Deep Ensemble | $0.30 \pm 0.09$ | $0.59 \pm 0.04$ | $0.50 \pm 0.13$ | $-14.3 \pm 4.7$ | $10.8 \pm 1.4$ | $3.4 \pm 0.6$ |
| DUE | $0.12 \pm 0.12$ | $0.15 \pm 0.03$ | $\mathbf{0.80 \pm 0.79}$ | $-13.0 \pm 0.52$ | $15.2 \pm 1.0$ | $3.5 \pm 0.1$ |
| DEUP | $\mathbf{0.47 \pm 0.03}$ | $0.63 \pm 0.05$ | $\mathbf{0.75 \pm 0.07}$ | $\mathbf{-3.5 \pm 0.25}$ | $\mathbf{36.1 \pm 2.5}$ | $\mathbf{13.1 \pm 0.9}$ |

Table 7: Drug combinations: quality of uncertainty estimates from different methods. *Corr. w. res.* shows correlation between model residuals and predicted uncertainties $\hat{\sigma}$. A best-case *Upper Bound* on *Corr. w. res.* is obtained from the correlation between $\hat{\sigma}$ and true samples from $\mathcal{N}(0, \hat{\sigma})$. *Ratio* is the ratio between col. 1 and 2 (larger is better). *Log-likelihood*: average over 3 seeds of per sample predictive log-likelihood. *Coverage Probability*: Percentage of test samples which are covered by the 68% confidence interval. *CI width*: width of the 86% confidence interval.

For each model, we report the per sample predictive log-likelihood, coverage probability and confidence interval width, averaged over 3 seeds.

We also computed the correlation between the residuals of the model $|\hat{\mu}(x_i) - y_i|$ and the predicted uncertainties $\hat{\sigma}(x_i)$. We noted that the different uncertainty estimation methods lead to different distributions $p(\hat{\sigma}(x))$. For example, predicted uncertainties obtained with DUE always have a similar magnitude. By contrast, DEUP yields a wide range of different predicted uncertainties.

These differences between the distributions $p(\hat{\sigma}(x))$ obtained with the different methods may have an impact on the correlation metric, possibly biasing the comparison of the different methods. In order to account for differences in the distribution $p(\hat{\sigma}(x))$ across methods, we report another metric which is the ratio between the observed correlation $Corr(|\hat{\mu}(x) - y|, \hat{\sigma}(x))$ and the maximum achievable correlation given a specific distribution $p(\hat{\sigma}(x))$.

This maximum achievable correlation (refered to as the *upper bound*) is not *per se* a comparison metric, and is estimated (given a specific $p(\hat{\sigma}(x))$) as follows: we assume that, for each example $(x_i, y_i)$, the predictive distribution of the model $\mathcal{N}(\hat{\mu}(x_i), \hat{\sigma}(x_i))$ corresponds exactly to the distribution of the target, *i.e.* $y_i \sim \mathcal{N}(\hat{\mu}(x_i), \hat{\sigma}(x_i))$. Under this assumption, the residual of the mean predictor follows a distribution $\mathcal{N}(0, \hat{\sigma}(x_i))$. We can then estimate the upper bound by computing the correlation between the predicted uncertainties $\hat{\sigma}(x_i)$ and samples from the corresponding Gaussians $\mathcal{N}(0, \hat{\sigma}(x_i))$. 5 samples were drawn from each Gaussian for our evaluation. This upper bound is reported in the Table.

Finally, we reported our comparison metric: the ratio between the correlation $Corr(|\hat{\mu}(x) - y|, \hat{\sigma}(x))$ and the upper bound. The higher the ratio is, the closer the observed correlation is to the estimated upper bound and the better the method is doing.

It is interesting to note that the upper bound is much lower for DUE compared to other methods, as its predicted uncertainties lie within a short range of values.

Predicted $\hat{\mu}$ and uncertainty estimates can be visualized in Figure 10 for different models. MC-dropout, Ensemble and DUE consistently underestimate uncertainty, while the out-of-sample uncertainties predicted by DEUP are much more consistent with the order of magnitude of the residuals. Moreover, we observed that DUE predicted very similar uncertainties for all samples, resulting in a lower upper-bound for the correlation between residuals and predicted uncertainties compared to other methods. We observed a similar pattern when experimenting with the other kernels available in the DUE package, including the standard Gaussian kernel.

Finally, we note that in the context of drug combination experiments, aleatoric uncertainty could be estimated by having access to replicates of a given experiment (*c.f.* Section D), allowing us to subtract the aleatoric part from the out-of-sample uncertainty, leaving us with the epistemic uncertainty only.

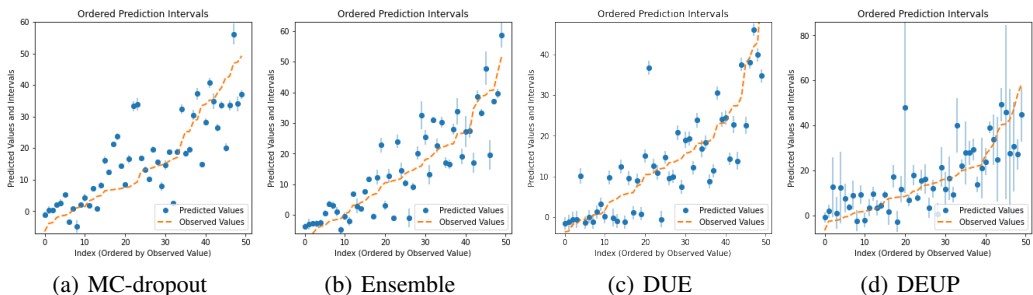

(a) MC-dropout      (b) Ensemble      (c) DUE      (d) DEUP

Figure 10: Predicted mean and uncertainty for different models on a separate test set. 50 examples from the test set are ordered by increasing value of true synergy score (orange). Model predictions and uncertainties are visualized in blue. MC-Dropout, Ensemble and DUE consistently underestimate the uncertainty while DEUP seems to capture the right order of magnitude. Figures made using The Uncertainty Toolbox (Chung et al., 2020).

One complete training run for the drug combination experiments takes about 0.01 GPU days on a V100 GPU. In total these set of experiments took about 0.2 GPU days on a Nvidia V100 GPU.

