# OpenReview forum: "DEUP: Direct Epistemic Uncertainty Prediction"
_ICLR.cc/2022/Conference — ICLR 2022 Submitted_

### Official Review · Reviewer_Scim · 2021-10-30

**Correctness:** 2
**Technical Novelty And Significance:** 2
**Empirical Novelty And Significance:** 2
**Recommendation:** 5
**Confidence:** 3

**Details Of Ethics Concerns:**

Uncertainty quantification is important because it helps represent what we do not know. This paper is not carefully reasoned and does not provide justifications on why one should use their definition or methods.

**Main Review:**

It's possible that there are some interesting ideas in this paper, but at this stage, I find it difficult to understand this paper.

The paper argues a different definition of epistemic uncertainty for the interactive learning settings but did not provide a clear argument on why their definition is more compelling than the existing ones. I suggest discussing how interactive learning is different from the standard setting, then motivating their definition.

The authors also seem to use out-of-sample generalization and out-of-distribution generalization interchangeability. For example, on page 1, it says ``The total expected out-of-sample loss at a point can be decomposed into an epistemic (reducible) part and an aleatoric (irreducible) part". Then on page 2, it says ``DEUP error predictors can be explicitly trained to care about, and calibrate for estimating the generalization error for examples which may come from a distribution different from the distribution of most of the training examples, i.e., an out-of-distribution (OOD) setting". out of sample generalization is about generalizing to the same distribution. Out of distribution generalization is about how to generalize to a different one. These are two very different concepts.

It's unclear what contributions 3  and 4 are supposed to mean.

Proposition 1 is not a result. It should not be a proposition. Is it supposed to be an illustrative example? The paragraph after proposition 1 is half-page long. I am guessing the goal is to discuss some mathematical insights? But it is impossible to parse.

The algorithm itself is also difficult to understand not at all self-contained. What is the stopping criterion? What does the ``optional (or every few interactions only)" statement supposed to mean? What is x_{acq}?  If a and e are estimators, what does a-e mean?
This is one of the main results of the paper. Yet it is not at all self-contained. A reader should not have to go to the appendix to understand the algorithm.



**Summary Of The Paper:**

This paper uses out-of-sample prediction error as a measurement of epistemic uncertainty.
Using this definition, it develops an estimator: direct epistemic uncertainty prediction.
The idea is to have a main predictor to learn the task, and an error predictor to predict the generalization error.
Empirical studies show that their proposed estimator produces better estimation on downstream tasks such as sequential model optimization and reinforcement learning.

**Summary Of The Review:**

It is possible that there are some interesting ideas, but the writing is extremely unclear. I cannot understand this paper.

---

> ### Author Response · Authors · 2021-11-14
> **Response to reviewer (1/2)**
>
> We thank the reviewer for the questions and comments!
>
> > The paper argues a different definition of epistemic uncertainty for the interactive learning settings but did not provide a clear argument on why their definition is more compelling than the existing ones. I suggest discussing how interactive learning is different from the standard setting, then motivating their definition.
>
> The interactive setting provides us with an oracle from which aleatoric uncertainty can be estimated. If we care about uncertainty in the output (which is certainly the target for deep learning, by opposition to parameter identification), then it makes sense to decompose output error into noise (aleatoric uncertainty) and a reducible part which is a form of epistemic uncertainty. The interactive setting also matters because this epistemic uncertainty estimator tells us about the potential to reduce both bias and variance (without having to explicitly disentangle them): if it is large, then either bias or variance or both are large.
>
> > The authors also seem to use out-of-sample generalization and out-of-distribution generalization interchangeability.
>
> We really mean different things. In some settings, there is no change in distribution that is relevant and we talk about OOS errors. In interactive settings where the distribution may change, it makes sense instead to talk about OOD generalization. In terms of theoretical analysis, we focus on the OOS but unchanging distribution case for the obvious reason that this is a setting where classical learning theory can be applied (and a corresponding framework for OOD settings is still lacking!)
>
> > For example, on page 1, it says The total expected out-of-sample loss at a point can be decomposed into an epistemic (reducible) part and an aleatoric (irreducible) part".
>
> By expected out-of-sample loss we actually mean the conventional loss in Machine Learning: $\mathbb{E}_{y \sim P(y|x)} [l(y, \hat{f}(x)]$
>
> >   Then on page 2, it says DEUP error predictors can be explicitly trained to care about, and calibrate for estimating the generalization error for examples which may come from a distribution different from the distribution of most of the training examples, i.e., an out-of-distribution (OOD) setting". out of sample generalization is about generalizing to the same distribution. Out of distribution generalization is about how to generalize to a different one. These are two very different concepts.
>
> We agree and are very conscious of the distinction. The point of this paragraph in Page 2 is to highlight an important feature of DEUP: If the error predictor is trained with features that are informative of whether the inputs are in distribution or not (e.g. density or variance), then using a suitable training set would lead to an error estimator with OOD generalization abilities. We will clarify as needed in the text.
>
> > It's unclear what contributions 3 and 4 are supposed to mean.
>
> The non-stationarity of the data to be fit by the secondary (DEUP) predictor (and how to mitigate that) is what contribution 3 is about and is discussed in section 3.2. Contribution 4 is about the positive comparative experimental results presented in section 5.
>
> > Proposition 1 is not a result. It should not be a proposition. Is it supposed to be an illustrative example?
>
> It is a mathematical result (proven in appendix) which shows, in the case of regression with Normally distributed noise (a fairly large and important class of problems), how the total error of a predictor can be decomposed into the proposed epistemic uncertainty and the additive noise level. Why do you say it is not a result?
>
> > The paragraph after proposition 1 is half-page long. I am guessing the goal is to discuss some mathematical insights? But it is impossible to parse.
>
> We are sorry that it was difficult to parse. The paper is already quite compressed and we put figure 2 in order to help grasp what is going on at a more visual and intuitive level. One solution would be to expand the explanation in the appendix. Would that make sense to you?
>
> > The algorithm itself is also difficult to understand not at all self-contained. What is the stopping criterion?
>
> The usual in standard iterative ML (like when training a neural net), e.g., your favorite early stopping method, or if a certain budget of oracle calls is reached.

---

> > ### Author Response · Authors · 2021-11-14
> > **Response to reviewer (2/2)**
> >
> > > What does the "optional (or every few interactions only)" statement supposed to mean?
> >
> > This sentence relates to training the aleatoric uncertainty estimator. Although in all our experiments in the main text, aleatoric uncertainty is zero, the proposed approach works when an estimator of aleatoric uncertainty is available. Such an estimator can be improved as more data is gathered. The corresponding line in the algorithm provides flexibility in terms of how often this estimator is fine-tuned, which might be computationally expensive. For instance, if aleatoric uncertainty is known to be almost constant for all regions in the input space, a reasonable estimate thereof can be obtained reasonably quickly in the first few iterations, and it might not be sound to spend extra computational resources at each subsequent iteration.
> >
> >
> > > What is $x_{acq}$?
> >
> > $x_{acq}$ is the batch of newly acquired data points, selected using an acquisition function based on the uncertainty of the predictor $f$.
> >
> > > If a and e are estimators, what does a-e mean?
> >
> > $e-a$ is the difference between the outputs of the estimators $e$ (total uncertainty) and $a$ (aleatoric uncertainty), which gives us an estimate of the epistemic uncertainty, as described in Definition 2.3. We will make this more clear in the algorithm.
> >
> > > This is one of the main results of the paper. Yet it is not at all self-contained. A reader should not have to go to the appendix to understand the algorithm.
> >
> > We have made efforts to ensure the algorithm is self-sufficient along with the main text, except for the estimation of aleatoric uncertainty, with most things discussed in section 3. We would be happy to improve the clarity based on any more suggestions you have!
> >
> > We would also like to address the ethical concerns raised by the reviewer:
> >
> > > This paper is not carefully reasoned and does not provide justifications on why one should use their definition or methods.
> >
> > We do not agree with the reviewer's assessment of the ethical risks posed by this work. We carefully proceed through the definitions and propositions and provide several motivations for the work along with various empirical results. Based on the reviewers' feedback we are working to make the paper more clear and accessible and are happy to address any specific clarifications from the reviewer.

---

> > > ### Comment · Reviewer_Scim · 2021-11-25
> > > **Thanks for the response.**
> > >
> > > I'd like to thank the authors for the updated draft and detailed responses. I've re-read the paper and the discussions. The updated manuscript is much clearer. I will update my score to reflect that.
> > >
> > > I largely agree with Reviewer 4Lh6's comments, which have been mostly addressed. However, I am still leaning towards a weak reject. Because this paper proposes a metric that could potentially be applied to evaluate an important line of work, it warrants more scrutiny. What are the potential drawbacks of using the generation error as a metric for uncertainty quantification? The authors wrote: the error predictor cannot generalize to places where the main predictor cannot generalize but can provide clues. Does this always hold true? When will the error predictor not provide the correct "clues"? A detailed discussion on the potential failure cases would be very beneficial.
> > >
> > > Minor comments & typos:
> > > - at the bottom of page 5, there is an extra parenthesis.
> > > - in algorithm 1, ... proposes new input points x_{acq} ...
> > > - since e and a are estimators instead of estimates,  the acq function should take e(x) - a(x) as input.

---

> > > > ### Author Response · Authors · 2021-11-26
> > > > **Thank you for the comments and suggestion**
> > > >
> > > > Indeed the idea of using the generalization error as a measure of epistemic uncertainty is novel. The drawback of the method, compared to existing methods, is that it doesn’t provide an explicit predictive distribution that reflects the learner’s subjective belief about the true value they are estimating; but it is a price to pay in order to have a metric that arguably captures the full extent of the lack of knowledge of the learner, which existing methods only try to give a proxy of.
> > > >
> > > > We would like to highlight however that it is not the error predictor that provides any clues whatsoever, but rather it is the stationarizing features we introduced in section 3.2 (such as the variance or the density) that provide clues to the error estimator.
> > > >
> > > > A simple way to visualize this is to consider what happens for an input x that’s very far from all training points seen so far by the main predictor:
> > > >
> > > > 1- such a point x is arguably “out-of-distribution”, and because it is very far, then a good density model or variance-based method such as a GP would estimate a low density and high variance for this point.
> > > >
> > > > 2- if the main predictor makes a mistake in predicting the value at x, which is likely given that x is OOD, then those variance and density estimates become clues to the secondary model that “low density + high variance” imply “high generalization error”.
> > > >
> > > > 3- Having multiple such x’s, for example after a few rounds of active learning or sequential model optimization, makes it possible for a secondary model with good generalization capabilities to directly predict whenever a new point with low density and high variance comes in, that there is a great generalization error (i.e. epistemic uncertainty). This is exactly what we meant when we wrote “The error predictor (without density or variance features) cannot generalize to places where the main predictor cannot generalize. However, the density and variance features can provide clues to the error predictor when it sees an out-of-distribution example, thus helping it provide sufficiently accurate estimates of uncertainty.”

---

> ### Author Response · Authors · 2021-11-24
> **Follow up**
>
> We thank the reviewer once again for the thoughtful review! As we approach the end of the discussion period, we would like to ask if the rebuttal and updated draft address all your questions and comments. We would be happy to discuss if you have any further questions or comments!

---

### Official Review · Reviewer_4Lh6 · 2021-10-31

**Correctness:** 2
**Technical Novelty And Significance:** 3
**Empirical Novelty And Significance:** 3
**Recommendation:** 6
**Confidence:** 3

**Main Review:**


This paper provides a relatively novel approach to uncertainty estimation, which explicitly includes estimation of model bias.  With a few exceptions, I found the paper to be well written. I found it to be an enjoyable read, as it made me think about and question some of the assumptions traditionally made in the deep learning uncertainty estimation literature. For this, I think that this work can present a relevant contribution to the ICLR community. Having said this, I have some doubts/concerns which I outline below:

**Major doubts**

1. The paper proposes to redefine uncertainty in terms of generalisation error instead of predictive variance. I am not sure how to feel about this:

Pros:

* The definition is clearly stated. The authors clearly put effort into avoiding confusion regarding this point. Additionally, the new definition clearly motivates the proposed method: a sensible alternative to existing methods.
*  The authors provide clear intuition for this definition in proposition 1, relating their definition of uncertainty to more traditional estimates of uncertainty like variance for specific choices of data generative process.

Cons:
  * This redefinition could cause confusion in the deep learning uncertainty quantification field, which is an already not very rigorous space.
  * The new definition is not always intuitive or does it provide interpretable uncertainty estimates. I think this is due to there not being an explicit predictive distribution. Consider a binary disease diagnostic setting: The primary model outputs a probability of disease of 87%. The secondary model outputs a risk estimate of 4.7. What does this mean?

I look forward to discussing this point with the authors and other reviewers.


2.  The authors motivate their approach by contrasting the sub optimality of the Bayesian update under model misspecification with the asymptotic unbiasedness off large neural networks:

	* The GP shown in the first toy example of the paper clearly has a strong preference over some functions over others. However, the rest of the examples in the paper use NNs. As stated by the authors these are asymptotically unbiased. In part, this is what makes Bayesian deep learning so difficult. As a result, it is not clear to me why bias would be such a large concern in the deep learning setting.
	* In section 3.1, the authors claim that DEUP has asymptotically optimal properties (for large enough secondary models and held out sets). It is not clear to me how this is a relevant property: under this setting the main model would have no uncertainty and we would not need an error predictor. In the finite data setting, if the error predictor incorporates less assumptions than Bayesian Methods it will have larger variance. I dont think it is clear enough from the paper that there is no free lunch in this regard.

As a result, I think that the claim made at the end of section 2 is unfounded: “directly estimating the epistemic uncertainty as the reducible part of the loss is superior to existing variance-based predictions of epistemic uncertainty.” I think the claim in the introduction is much more reasonable ”direct estimation of epistemic uncertainty can lead to more precise estimates than variance-based ones.” I encourage the authors to explicitly discus the above and relax their claims.

3. I found section 3.2 to not be very clear. The authors state that variance-based uncertainty estimators can not deal with non-stationarity in the observation distribution arising from an active learning setting. It is not clear to me why this is the case: as more data is observed, independently of what distribution it comes from, the range of possible functions that fit all the datapoints decreases and so does the variance. What am I missing here? In fact, I found this section to be written in a somewhat confusing manner overall. I am not sure I understood the problem. I encourage the authors to re-write this section.



**Minor concerns**


1. At the end of the introduction, the authors mention dropout and ensembles as approaches to approximate the Bayesian predictive variance.  While the former is a rather crude approximation, the latter does not target the Bayesian posterior at all. Perhaps HMC is a better example.

2. It took some effort to understand Fig 2 because it kind of looks like the grey area corresponds to posterior variance while orange is bias. In fact, the bias and variance labels are for the straight and dashed arrows. Does it make sense to have this diagram? Is it needed to understand the rest of the paper? Perhaps It would be more useful if it were made more relevant to the proposed method DEUP: for example, the authors could state how DEUP targets \epsilon instead of \Delta in the plot.


**Experiments**


1. The proposed method is somewhat elaborate: It combines a k-fold error estimation, with a NN variance estimator (DUQ specifically) and a deep generative model to produce uncertainty estimates. It is somewhat unsurprising that this combination is able to outperform a variance estimator on its own.

2. Apart from the toy GP experiment, I do not think that any of the experiments of the paper target the main claim of DEUP capturing error from bias in addition to variance.

3. On the GP experiment: As the authors state, the GP is not allowed access to the held out data, while DEUP is. I understand this choice was made to drive home a point. However, it would have been interesting to see the setting in which the GP does have access to this data, as this setting better matches that of the rest of the paper. The Bayesian marginal likelihood could also be seen as a proxy for generalisation error and it would be interesting to see it compared against DEUP.

4. In the image classification OOD rejection experiment, it is not obvious to me why we see such a difference in results when using Spearman-rank correlation and AUROC as metrics. Do the authors have any intuition for that?

5. The remaining are very cool but non-standard for this type of paper. It's hard to say much about them as I am not familiar with the settings.
	* Their diversity is a definite plus
	* They seem to be relatively toy tasks however. I am somewhat surprised by this as I would have guessed the neural-network based approach to uncertainty estimation allows for good scalability.

6.  Why is the vanilla DQN baseline better than uncertainty aware methods in Fig 5?

**Summary Of The Paper:**

Given some supervised task, this paper redefines the uncertainty of a solution as its generalisation risk. The authors then propose to learn a secondary function to estimate the generalisation risk of the first. This is done by using held-out points as training data for the secondary model. In turn, this held out used to estimate the primary model error comes from a k-fold split.  The inputs to the secondary model are the data points being evaluated, estimates of the predictive variance of the primary model, density estimates from a generative model and whether the point has been observed by the primary model.

The authors posit that the advantage of their method is that it can capture uncertainty due to model selection bias, something omitted by existing work, which focuses on the variance of the learnt estimator. The authors provide a diverse range of experiments: OOD rejection in image classification, active learning for drug discovery, function optimisation and exploration in reinforcement learning.

**Summary Of The Review:**

This paper provides a novel view on uncertainty estimation and is an interesting read. I believe it is held back by some overly ambitions claims and a lack of depth in discussion at some points. Experimentally the paper is again interesting but fails to drive home its main points.

I currently can not recommend acceptance. If the authors improve their discussion and claims I will raise my score to a weak accept. If the authors provide some experimental evidence for the existence of bias in a realistic setting and DEUP accurately estimating that bias while variance-based methods to fail, I will raise my score to a strong accept.

**update**

After a discussion with the authors I have raised my score from 5 to 6.

---

> ### Author Response · Authors · 2021-11-14
> **Response to reviewer (1/4)**
>
> We thank the reviewer for the thoughtful comments and questions!
>
> > This redefinition could cause confusion in the deep learning uncertainty quantification field, which is an already not very rigorous space.
>
> That is why the paper carefully proceeds via definitions and propositions. The general idea is simple, though, and illustrated in figure 2: traditional variance-based methods may be missing an aspect of epistemic uncertainty, that we define to be the lack of knowledge of the learner, due to bias (misspecification). DEUP opens the door to methods that may be robust to this. Instead of thinking only about the uncertainty regarding which function under the prior is correct, DEUP also considers the possibility that none of them is correct. It focuses on the lack of knowledge about the correct conditional expectation at specific points (rather than some parameters of a model) which makes sense for things like deep nets (for which there is a many-to-one mapping of parameters to function), especially given that it is this lack of knowledge that should drive acquisition in active learning or exploration in reinforcement learning.
>
> > The new definition is not always intuitive or does it provide interpretable uncertainty estimates. I think this is due to there not being an explicit predictive distribution.
>
> As constructed, DEUP does not provide an explicit predictive distribution, but only estimates of its first and second moments. However, as mentioned in a previous reply, one could potentially train a DEUP network to predict the quantiles of the deviation to the mean prediction to the truth, using a quantile regression loss.
>
> > Consider a binary disease diagnostic setting: The primary model outputs a probability of disease of 87\%. The secondary model outputs a risk estimate of 4.7. What does this mean? I look forward to discussing this point with the authors and other reviewers.
>
> We consider two ways of answering your question below.
>
> (1) First of all, we focus on the regression problem. So let us say the primary model outputs the conditional expectation of a real variable (like a patient wellness score). Let us say it predicts a value of 0.87. Now let us say the secondary model outputs 0.01. What this means is that we expect the root mean squared error around the 0.87 to be sqrt(0.01)=0.1. Ignoring higher moments, this corresponds to a Gaussian whose mean is 0.87 (for this patient) and standard deviation is 0.1.
>
> (2) Second, similar to when humans use assessments of uncertainty in everyday language, instead of focusing on the raw uncertainty estimate provided by DEUP for a single data point, we can compare two risk estimates at two different data points. Regardless of the output of the primary model, if the uncertainty predicted by the secondary model at a point $x_1$ is greater than that predicted at $x_2$, this indicates that a budget-constrained learner needs to prioritize learning more at $x_1$ (or near $x_1$) over $x_2$.

---

> > ### Author Response · Authors · 2021-11-14
> > **Response to reviewer (2/4)**
> >
> > > The authors motivate their approach by contrasting the sub optimality of the Bayesian update under model misspecification with the asymptotic unbiasedness off large neural networks:
> > The GP shown in the first toy example of the paper clearly has a strong preference over some functions over others. However, the rest of the examples in the paper use NNs. As stated by the authors these are asymptotically unbiased. In part, this is what makes Bayesian deep learning so difficult. As a result, it is not clear to me why bias would be such a large concern in the deep learning setting.
> >
> > That is a good point, but one has to remember that typical large neural nets, as well as GPs, have hyper-parameters (or training procedures) that regularize or smoothen them to avoid overfitting. If the training set is not very large, we end up regularizing a lot. In the case of large neural nets, there is evidence that a form of auto-regularization is taking place by the SGD training procedure itself (and even more so if early stopping is used). In other words, on a small training set, a large neural net will not exploit all its capacity (either because of SGD, regularization, early stopping or all of these), meaning that it will incur a bias: the set of functions it focuses on is e.g. overly smooth (in the case of GPs this is very certainly what happens). Hence as the training set size increases, that bias will decrease, meaning that it may be more worthwhile acquiring these data points where the reduction in bias would be larger. Keep in mind that in this paper we are motivated by active learning (or black-box optimization) settings where the learner uses epistemic uncertainty estimation in order to decide what examples to label next.
> > Additionally, we would like to highlight that the experiments in Section 5.2 do not use a NN as a primary model, but actually a GP for fair comparisons with state-of-the-art GP-based methods. This flexibility of what type of primary (or secondary) model to use is a feature of the proposed approach.
> >
> > > In section 3.1, the authors claim that DEUP has asymptotically optimal properties (for large enough secondary models and held out sets). It is not clear to me how this is a relevant property: under this setting the main model would have no uncertainty and we would not need an error predictor. In the finite data setting, if the error predictor incorporates less assumptions than Bayesian Methods it will have larger variance.
> >
> > It is indeed true that at the limit of infinite data, there would be no uncertainty, and having a secondary model would be pointless. The sole purpose of this paragraph of Section 3.1 is to highlight the soundness of training the error predictor with the specified targets: given this dataset of input-target pairs, what would the resulting predictor be an estimator of? A natural way to answer this question is to verify what happens at the (impossible) limit of infinite data. We will clarify that the only setting we care about is the realistic finite data one, and that the reasoning at the infinite data limit is for theoretical soundness only.
> >
> > > I dont think it is clear enough from the paper that there is no free lunch in this regard.
> >
> > We did not claim any free lunch. However, there may be some opportunity to identify examples with high epistemic uncertainty in ways that are quite different from the different approaches, for the reasons explained and as demonstrated experimentally, that can bring an advantage. We are particularly excited about the active learning setting both because of our results and because aleatoric uncertainty can be estimated there (making our approach also more applicable in theory).
> >
> > > As a result, I think that the claim made at the end of section 2 is unfounded: “directly estimating the epistemic uncertainty as the reducible part of the loss is superior to existing variance-based predictions of epistemic uncertainty.” I think the claim in the introduction is much more reasonable ”direct estimation of epistemic uncertainty can lead to more precise estimates than variance-based ones.” I encourage the authors to explicitly discus the above and relax their claims.
> >
> > We will use the formulation in the introduction, as we do not wish to make the claim that the proposed approach is necessarily superior in all settings. Thanks for catching that.

---

> > > ### Author Response · Authors · 2021-11-14
> > > **Response to reviewer (3/4)**
> > >
> > > > I found section 3.2 to not be very clear. The authors state that variance-based uncertainty estimators can not deal with non-stationarity in the observation distribution arising from an active learning setting. It is not clear to me why this is the case: as more data is observed, independently of what distribution it comes from, the range of possible functions that fit all the datapoints decreases and so does the variance. What am I missing here? In fact, I found this section to be written in a somewhat confusing manner overall. I am not sure I understood the problem. I encourage the authors to re-write this section.
> > >
> > > Thanks for pointing out that the section needs more clarity. The basic argument is that in active learning the stream of examples is not iid. Indeed the variance and uncertainty will decrease with more data, but the question we focus on is whether these variance-based uncertainty estimates could be less reliable because of the distributional shift and if this could be mitigated. For example, if there is a systematically larger error on new batches of examples (because of the distributional shift), this will be immediately reflected in the DEUP predictor (since the measured new-batch errors will be larger than would have been in-distribution errors). The section builds upon two facts: 1- the error is a function of both the input $x$ and the training dataset $\mathcal{D}$, and 2- we cannot provide $\mathcal{D}$ as an input to the error predictor, and thus we propose to replace it with the stationarizing features.
> > >
> > >
> > > > At the end of the introduction, the authors mention dropout and ensembles as approaches to approximate the Bayesian predictive variance. While the former is a rather crude approximation, the latter does not target the Bayesian posterior at all. Perhaps HMC is a better example.
> > >
> > > We are not claiming ensembles are doing a great job from a Bayesian perspective, but there is a strong relationship between each chain (randomly initialized in a different starting parameter vector) in a Langevin MCMC (a la Teh \& Welling 2011), which is basically a form of SGD, and SGD training of each network in an ensemble.
> > >
> > > > It took some effort to understand Fig 2 because it kind of looks like the grey area corresponds to posterior variance
> > >
> > > Yes
> > >
> > > > while orange is bias.
> > >
> > > No b = f* - f, as indicated in the caption. Orange is the part of the prior support region with low probability under the posterior.
> > >
> > > > In fact, the bias and variance labels are for the straight and dashed arrows.
> > >
> > > Exactly.
> > >
> > > > Does it make sense to have this diagram? Is it needed to understand the rest of the paper?
> > >
> > > It illustrates the role of bias (misspecification) and the difference between estimating the variance $E[(\hat{f} - f)^2]$ and predicting the total lack of knowledge $(f^* - \hat{f})^2$.
> > >
> > > > Perhaps It would be more useful if it were made more relevant to the proposed method DEUP: for example, the authors could state how DEUP targets $\epsilon$ instead of $\Delta$ in the plot.
> > >
> > > Indeed. We will do that. Thanks.
> > >
> > > > The proposed method is somewhat elaborate: It combines a k-fold error estimation, with a NN variance estimator (DUQ specifically) and a deep generative model to produce uncertainty estimates. It is somewhat unsurprising that this combination is able to outperform a variance estimator on its own.
> > >
> > > It is true that the approach requires extra steps, but all of them are necessary to obtain good estimators of the lack of knowledge of the learner, which is what active learning is all about.
> > >
> > > > Apart from the toy GP experiment, I do not think that any of the experiments of the paper target the main claim of DEUP capturing error from bias in addition to variance.
> > >
> > > This is a good point, and we thought about it, but unless we know the ground truth $f^*$ (as in the toy experiments we performed), it is difficult to control for bias! Note however that in both active learning experiments (RL and Sequential Optimization), DEUP capturing the bias is the only reason its uncertainty estimates, used by the acquisition function, lead to faster learning curves.
> > >
> > > > On the GP experiment: As the authors state, the GP is not allowed access to the held out data, while DEUP is. I understand this choice was made to drive home a point. However, it would have been interesting to see the setting in which the GP does have access to this data, as this setting better matches that of the rest of the paper.
> > >
> > > When we give the GP access to the data it learns to predict the correct output accurately for the point and has very low uncertainty there, as one would expect. The main thing that distinguishes DEUP is that it can leverage the OOD data which arises naturally in an interactive learning setting to improve the uncertainty estimates for OOD data.

---

> > > > ### Author Response · Authors · 2021-11-14
> > > > **Response to reviewer (4/4)**
> > > >
> > > > >   The Bayesian marginal likelihood could also be seen as a proxy for generalisation error and it would be interesting to see it compared against DEUP.
> > > >
> > > > Right, and the difference, in average, should tell us something about the bias (but would also compound sources of error either in estimating the Bayesian marginal likelihood or in training DEUP).
> > > >
> > > > > In the image classification OOD rejection experiment, it is not obvious to me why we see such a difference in results when using Spearman-rank correlation and AUROC as metrics. Do the authors have any intuition for that?
> > > >
> > > > One of the motivations for using the Spearman-rank correlation over AUC was that it allows us to get a better view of how well the estimates actually correlate with the underlying true epistemic uncertainty. The AUC only captures whether a method can estimate uncertainties well enough to detect OOD samples, but does not capture the correlation between the estimate and the true value of the epistemic uncertainty.
> > > >
> > > > > The remaining are very cool but non-standard for this type of paper. It's hard to say much about them as I am not familiar with the settings. They seem to be relatively toy tasks however. I am somewhat surprised by this as I would have guessed the neural-network based approach to uncertainty estimation allows for good scalability.
> > > >
> > > > This is certainly a motivation for us. One has to start by walking before running, and our expertise and compute power is better suited to the development of new algorithms at that scale. We are hoping these good results will encourage others to do the required engineering to develop large-scale versions of DEUP. Note that large number of experiments we performed is already quite a significant computational and experimental burden and provides strong clues about the usefulness of the method supporting the theoretical arguments about bias. The functions used (up to 100 dimensions) in Section 5.2 are actually common benchmarks in the Sequential Optimization literature.
> > > >
> > > > > Why is the vanilla DQN baseline better than uncertainty aware methods in Fig 5?
> > > >
> > > > Our observations match the results from Osband et al 2020, where the vanilla DQN performs better than uncertainty aware methods like Bootstrap-DQN. One of the hypothesis we have is that these methods might be prone to getting stuck in suboptimal policies due to inefficient exploration in a relatively simple environment, whereas the e-greedy exploration in DQN is enough to just solve the task. We believe this requires further extensive testing on a broader set of environments.

---

### Official Review · Reviewer_zaec · 2021-11-01

**Correctness:** 4
**Technical Novelty And Significance:** 3
**Empirical Novelty And Significance:** 3
**Recommendation:** 5
**Confidence:** 4

**Main Review:**

Strengths of the paper:

- Attempts to estimate the epistemic uncertainty in a creative way (total minus aleatoric).
- Methods cover both static and interactive use cases
-

Weaknesses of the paper:

- Dependence on the Oracle for estimating Aleatoric uncertainty makes its practical use very limited IMO. While the approach bypasses some of the limitations of existing epistemic uncertainty techniques (e.g vulnerability to misspecification), the dependence on having an aleatoric uncertainty estimator or Oracle can be a major limitation as well.

- The generalization of the secondary model (the one that predicts total generalization error) itself feels to be on shaky grounds. It appears that the main difference in the secondary model (compared to similar works that employ a secondary model) is that it uses labels from out-of-sample examples. However those examples are (by definition) something already available at training time, and it's not clear how or why we somehow ended up with the ability to predict total error in OOD settings?

- The overall narrative / presentation is hard to follow. There is too much of back and forth between the main paper and Appendix. Some of the key elements of the paper (e.g Algorithm 2, which is likely of high interest to a majority of readers) is down in the appendix.

**Summary Of The Paper:**

This paper proposes a method to estimate the epistemic uncertainty (uncertainty due to lack of knowledge/data) at a new model input. The paper takes an indirect approach towards this goal by a) first estimate the generalization error at the new input, b) next estimate the aleatoric error (inherent uncertainty in the data distribution / irreducible error), and c) subtract aleatoric error estimate from the total generalization error estimate to obtain the epistemic uncertainty estimate.

The authors claim that this method captures both the uncertainty due to lack of data, as well as model misspecification in the process - while other/existing methods focus mostly on the variance of the posterior distribution (or its approximation) as a measure of epistemic uncertainty (and thereby implicitly assume the model is well specified.)

Estimating the total generalization error itself is performed with a second neural network model, which uses residuals obtained from the primary model as its labels. Estimating aleatoric error either assumes the presence of an Oracle or

The authors apply their technique on both static (fixed data set) as well as interactive (active learning) settings, though mostly focused on the mean squared error loss function.

**Summary Of The Review:**

The main factors influencing the recommendation are the strengths and weaknesses highlighted above.

The paper can be improved significantly by:

a) Adding narrative to help justify why the secondary network generalized to OOD settings where the primary network couldn't. Perhaps present results that shine light on just this sub-aspect.

b) Improving the overall flow and presentation.

---

> ### Author Response · Authors · 2021-11-14
> **Response to reviewer**
>
> We thank the reviewer for the thoughtful comments!
>
> > Dependence on the Oracle for estimating Aleatoric uncertainty makes its practical use very limited IMO. While the approach bypasses some of the limitations of existing epistemic uncertainty techniques (e.g vulnerability to misspecification), the dependence on having an aleatoric uncertainty estimator or Oracle can be a major limitation as well.
>
> Indeed requiring access to an oracle could be a limitation depending on the application setting. However, as we demonstrate in our experiments, the aleatoric uncertainty can be ignored, while still getting useful estimates of the epistemic uncertainty. It is also interesting to note that there are practically important scenarios, like active learning, where it is possible to query the oracle twice on the same input, and where aleatoric uncertainty can thus be estimated, and DEUP can take advantage of that.
>
> > The generalization of the secondary model (the one that predicts total generalization error) itself feels to be on shaky grounds. It appears that the main difference in the secondary model (compared to similar works that employ a secondary model) is that it uses labels from out-of-sample examples. However those examples are (by definition) something already available at training time, and it's not clear how or why we somehow ended up with the ability to predict total error in OOD settings?
>
> It depends on the OOD setting we care about. In active learning, which has motivated this work, the OOD-ness arises, like in continual learning, because we get a stream of batches of examples whose distribution may shift. In that case, the learner has access to examples of that shift. If there is absolutely no information about the change in distribution, then no method can provide strong guarantees, even using density-based approaches. However, DEUP can take advantage of heuristics like density and epistemic uncertainty estimated from variance and combine them as features which may generalize in such difficult OOD settings, as we tried to explain in Section 3.2.
>
> > The overall narrative / presentation is hard to follow. There is too much of back and forth between the main paper and Appendix. Some of the key elements of the paper (e.g Algorithm 2, which is likely of high interest to a majority of readers) is down in the appendix.
>
> Algorithm 2, is just a sub-case of the more general Algorithm 1 (with no acquisition step). We have included most of the key results and insights in the main text, but thank you for pointing this out, we will try to improve the flow of the main text.

---

> ### Author Response · Authors · 2021-11-24
> **Follow up**
>
> We thank the reviewer once again for the thoughtful review! As we approach the end of the discussion period, we would like to ask if the rebuttal and updated draft address all your questions and comments. We would be happy to discuss if you have any further questions or comments!

---

> ### Author Response · Authors · 2021-11-29
> **Any further questions or clarifications**
>
> With the discussion period about to end, we hope the rebuttal and updated draft addressed all your questions and concerns, we would be happy to clarify further if you have any remaining questions or clarifications about the rebuttal or the updated draft!

---

### Official Review · Reviewer_iSGN · 2021-11-02

**Correctness:** 4
**Technical Novelty And Significance:** 4
**Empirical Novelty And Significance:** 2
**Recommendation:** 6
**Confidence:** 4

**Main Review:**

Pros:

1. The paper focuses on estimating epistemic uncertainty of machine learning models. Estimating predictive uncertainty of black box models is an important research area with promises on enabling risk aware decision making, which will enable safe deployment of such models to production.
2. The proposed approach is practical, which is important for its adoption by others. Interesting extensions are proposed (e.g. data density estimation and model variance features) that helps with improving DEUP's performance. It also does not affect the original model training, which is a positive.
3. Experiments are conducted in different settings with compelling performance compared to the state of the art methods like deep ensembles and MC Dropout.

Cons:

1. The paper compares DEUP's uncertainty estimates to variance estimates obtained from other techniques. Variance (and standard deviation) provides intuitive explanations on what they are capturing (i.e. the dispersion of predicted values). Furthermore, they have nice links to prediction intervals, which also provides nice intuitions on the estimated parameters with their probability. The other techniques (e.g. ensembles and MC Dropout) that are indicated as yielding variance estimates can also be directly used to compute such intervals from their empirical predictive distributions. I think the proposed uncertainty estimates would benefit a lot from such intuitions. For instance, by training the error predictor to predict the square error loss, my intuition is that DEUP is predicting extreme quantiles. I would love to hear authors' intuition as well as how the uncertainty estimates map to prediction intervals.
2. Since the approach relies on a predictive model to estimate the errors, do the authors believe that DEUP will be susceptible to issues predictive models face (e.g. how important is model fit, how to prevent overfitting)? For instance, in the case of fixed training set where the set used does not capture OOD dataset and density features are not used, is DEUP's error prediction able to generalize to OOD?
3. Questions on experiments:
    1. Section 5.1.2: I like the OOD analysis conducted. I have some minor comments:
        1. CIFAR-10 with SVHN provides a very strong OOD dataset, which is also evident from high AUROC values. In such a setting, I am interested to hear why the authors think the rank correlation metric between predicted uncertainty and OOD generalization error is a good metric. I actually would be interested to see an analysis where OOD definition is more subtle/gradual. For instance, Ovadia et al. 2019 provides such an analysis with increasing distribution shifts. I would be very interested to see how the method performs in such settings.
    2. Section 5.1.3: The metrics used here can be improved (except log likelihood). For instance, coverage probabilities and CI widths would be great metrics to use here, which will also help with the intuition point made above. I am not able to fully grasp what Upper Bound on Corr. w. res. metric is capturing (as well as the Ratio metric).

Minor comments:
- Algorithm 1 requires an estimator of aleatoric uncertainty. Does the authors use that in any of the experiments and how do they define it?
- Appendix pages 19, 20, 26 has missing links (indicated with ??'s).
- Appendix Table 6: I would be very interested to see the performance of features by themselves in this table.

**Summary Of The Paper:**

This paper proposes a new approach for computing epistemic uncertainty. The proposed approach, DEUP, builds a new model (in addition to the original model) which predicts epistemic uncertainty, defined as generalization error minus aleatory uncertainty. I highlight some features of DEUP below:
- DEUP works with a hold-out dataset that is used for training the error predictor.
- In case there does not exist a hold-out set or in interactive settings (like RL or active learning), DEUP is extended to be used in a cross-validation setting and the features used to fit the error predictor is extended to include data density estimates and model variance.
DEUP is evaluated on different settings including OOD data, sequential model optimization and RL.

**Summary Of The Review:**

Overall I vote for accepting the paper. I like the approach of directly estimating uncertainty. I have some concerns on providing intuition on the estimated uncertainty, which I believe could be addressed by some additional discussion and some more analyses. Hopefully the authors can address my concern in the rebuttal period.

---

> ### Author Response · Authors · 2021-11-14
> **Response to reviewer (1/2)**
>
> We thank the reviewer for the thoughtful comments and questions!
>
> > The paper compares DEUP's uncertainty estimates to variance estimates obtained from other techniques. Variance (and standard deviation) provides intuitive explanations on what they are capturing (i.e. the dispersion of predicted values). Furthermore, they have nice links to prediction intervals, which also provides nice intuitions on the estimated parameters with their probability. The other techniques (e.g. ensembles and MC Dropout) that are indicated as yielding variance estimates can also be directly used to compute such intervals from their empirical predictive distributions. I think the proposed uncertainty estimates would benefit a lot from such intuitions. For instance, by training the error predictor to predict the square error loss, my intuition is that DEUP is predicting extreme quantiles. I would love to hear authors' intuition as well as how the uncertainty estimates map to prediction intervals.
>
> The uncertainty estimates from DEUP can be directly interpreted as an estimate of the expected squared-error that the model will make on a given input. This corresponds to an estimate of the (conditional, point-wise) second moment of the output (the first one being given by the main predictor output itself). We have not attempted in the paper to interpret these estimates in terms of prediction intervals, but of course if one was willing to make a conditionally Gaussian assumption on these errors the predicted squared error would correspond to a conditional variance and one could calculate corresponding intervals. Alternatively and to avoid a parametric assumption like this, one could train the DEUP (secondary) predictor to predict chosen quantiles of the error, using a conditional quantile regression loss rather than squared error when predicting the difference between the main predictor $f(x)$ and the realization $Y$.
>
> > Since the approach relies on a predictive model to estimate the errors, do the authors believe that DEUP will be susceptible to issues predictive models face (e.g. how important is model fit, how to prevent overfitting)? For instance, in the case of fixed training set where the set used does not capture OOD dataset and density features are not used, is DEUP's error prediction able to generalize to OOD?
>
> The error estimator, like any other machine learning model, is susceptible to things like overfitting and needs proper regularization, early stopping, etc. Even without the density and variance features, DEUP gets a SRCC score of 0.36 (Appendix F), which is slightly lower than DUE, DUQ, and Ensembles but higher than Dropout. Our hypothesis, explained in Section 3.2, is that the variance and density estimators are important since they provide the error estimator clues which can help in generalizing the error estimates to OOD examples.
>
>
> > Section 5.1.2: I like the OOD analysis conducted. I have some minor comments. CIFAR-10 with SVHN provides a very strong OOD dataset, which is also evident from high AUROC values. In such a setting, I am interested to hear why the authors think the rank correlation metric between predicted uncertainty and OOD generalization error is a good metric. I actually would be interested to see an analysis where OOD definition is more subtle/gradual. For instance, Ovadia et al. 2019 provides such an analysis with increasing distribution shifts. I would be very interested to see how the method performs in such settings.
>
> The AUC metric used in such settings measures the ability of the uncertainty estimates to discriminate OOD examples. However, the estimates can be useful in detecting the OOD examples while not being correlated with the true epistemic uncertainty. The rank correlation between the predicted uncertainty and the (pointwise) generalization error allows us to measure "how accurate were the model's predictions of how wrong it is", which is a better metric in settings where you care about getting accurate uncertainty estimates. Thanks for the pointer to the distribution shift setup from Ovadia et al., 2019, we ran experiments on 5 of the 16 corruptions from CIFAR10C. You can find there results [here](https://ibb.co/vkcDXQw). We observe that while the SRCC is lower compared to experiments with completely OOD data (SVHN), DEUP still outperforms the baselines consistently. We will add the experiments on all the remaining corruptions (which we are currently running experiments on) in the updated draft.

---

> > ### Author Response · Authors · 2021-11-14
> > **Response to reviewer (2/2)**
> >
> > > Section 5.1.3: The metrics used here can be improved (except log likelihood). For instance, coverage probabilities and CI widths would be great metrics to use here, which will also help with the intuition point made above. I am not able to fully grasp what Upper Bound on Corr. w. res. metric is capturing (as well as the Ratio metric).
> >
> > For each model, we computed the correlation between the residuals of the model $|\hat{\mu}(x_i) - y_i|$ and the predicted uncertainties $\hat{\sigma}(x_i)$. We noted that the different uncertainty estimation methods lead to different distributions $p(\hat{\sigma}(x))$. For example, predicted uncertainties obtained with DUE always have a similar magnitude. By contrast, DEUP yields a wide range of different predicted uncertainties.
> >
> > These differences between the distributions $p(\hat{\sigma}(x))$ obtained with the different methods may have an impact on the correlation metric, possibly biasing the comparison of the different methods. In order to account for differences in the distribution $p(\hat{\sigma}(x))$ across methods, we report another metric which is the ratio between the observed correlation $Corr(|\hat{\mu}(x) - y|, \hat{\sigma}(x))$ and the maximum achievable correlation given a specific distribution $p(\hat{\sigma}(x))$.
> >
> > This maximum achievable correlation (refered to as the *upper bound*) is not *per se* a comparison metric, and is estimated (given a specific $p(\hat{\sigma}(x))$) as follows: we assume that, for each example $(x_i, y_i)$, the predictive distribution of the model $\mathcal{N}(\hat{\mu}(x_i), \hat{\sigma}(x_i))$ corresponds exactly to the distribution of the target, *i.e.* $y_i \sim \mathcal{N}(\hat{\mu}(x_i), \hat{\sigma}(x_i))$. Under this assumption, the residual of the mean predictor follows a distribution $\mathcal{N}(0, \hat{\sigma}(x_i))$. We can then estimate the upper bound by computing the correlation between the predicted uncertainties $\hat{\sigma}(x_i)$ and samples from the corresponding Gaussians $\mathcal{N}(0, \hat{\sigma}(x_i))$. 5 samples were drawn from each Gaussian for our evaluation. This upper bound is reported in the Table.
> >
> > Finally, we reported our comparison metric: the ratio between the correlation $Corr(|\hat{\mu}(x) - y|, \hat{\sigma}(x))$ and the upper bound. The higher the ratio is, the closer the observed correlation is to the estimated upper bound and the better the method is doing.
> >
> > It is interesting to note that the upper bound is much lower for DUE compared to other methods, as its predicted uncertainties lie within a short range of values.
> >
> > We will extend the explanations currently available in the Appendix of the paper, and add a pointer to them. Further, we will add the CI widths and coverage probabilities to our results in the updated draft.
> >
> >
> > > Algorithm 1 requires an estimator of aleatoric uncertainty. Does the authors use that in any of the experiments and how do they define it?
> >
> > We do not use the aleatoric uncertainty estimator, except for Figure 6 in the appendix. We ignore the effect of noise for all the other experiments.
> >
> > > Appendix pages 19, 20, 26 has missing links (indicated with ??'s).
> >
> > Thanks, we will fix these links.
> >
> > > Appendix Table 6: I would be very interested to see the performance of features by themselves in this table.
> >
> > This is already reported in the paragraph above the table at the end of Page 19. We will move these to the table for more clarity. Thanks.

---

> ### Author Response · Authors · 2021-11-24
> **Follow up**
>
> We thank the reviewer once again for the thoughtful review! As we approach the end of the discussion period, we would like to ask if the rebuttal and updated draft address all your questions and comments. We would be happy to discuss if you have any further questions or comments!

---

> ### Author Response · Authors · 2021-11-29
> **Further questions or clarifications**
>
> With the discussion period about to end, we hope the rebuttal and updated draft addressed all your questions and concerns, we would be happy to clarify further if you have any remaining questions or clarifications about the rebuttal itself or the updated draft!

---

### Author Response · Authors · 2021-11-14
**Overall response**

We thank all the reviewers for the thoughtful comments and questions. We have addressed the concerns of the reviewers and answered the questions in our individual responses to the reviewers. To address some common points:

1. **Generalization of the error predictor**:

The error predictor (without density or variance features) cannot generalize to places where the main predictor cannot generalize. However, the density and variance features can provide "clues" to the error predictor when it sees an out-of-distribution example, thus helping it provide sufficiently accurate estimates of uncertainty.

2. **Flow and Clarity**:

We have received several useful suggestions from the reviewers for improving the flow and clarity of the paper. We will make the discussion more clear and try to improve the presentation.

3. **Comparison to existing methods of modeling epistemic uncertainty**:

We will highlight that as its name suggests, DEUP tries to directly capture the lack of knowledge of the learner about the underlying ground truth conditional expectation, defined as the expected loss (minus aleatoric uncertainty); and that it is this lack of knowledge that is important in active learning scenarios, which is the motivation of the paper. This is in contrast to existing variance-based methods that provide only proxies of this lack of knowledge.


We will be updating the paper draft over the next few days. In the meantime, we look forward to a fruitful discussion with the reviewers!

---

### Public Comment · ~Meet_P._Vadera1 · 2021-11-16
**Interesting paper!**

Hi,

This is an interesting paper! I enjoyed reading it :)

We have some past work around distilling Bayesian posterior distribution for uncertainty estimation as well as a benchmark paper looking at various uncertainty quantification tasks (including OOD detection) that we think is relevant to this paper [1, 2]- hope you find it useful!

1. Meet P. Vadera, Brian Jalaian, and Benjamin M. Marlin. Generalized bayesian posterior expectation distillation for deep neural networks. In Proceedings of the 36th Conference on Uncertainty in Artificial Intelligence (UAI), volume 124 of Proceedings of Machine Learning Research, pages 719–728. PMLR, 03–06 Aug 2020b. URL https://proceedings.mlr.press/v124/ vadera20a.htm

2. Meet P. Vadera, Adam D. Cobb, B Jalaian, and Benjamin M. Marlin. URSABench: Comprehensive Benchmarking of Approximate Bayesian Inference Methods for Deep Neural Networks. In ICML Workshop on Uncertainty and Robustness in Deep Learning, 2020a

---

> ### Author Response · Authors · 2021-11-21
> **Related Work**
>
> Thank you for the kind words and the references! These are very interesting papers. We have added references to the papers and a short discussion in the related work section in our updated draft.
>
> To summarize the key differences here, as we discuss in the paper, just the posterior distribution variance (the standard way of getting uncertainty estimates from a Bayesian posterior) can be a bad proxy for the "lack of knowledge" of a learner, since it does include the effect of model misspecification. DEUP attempts to address this issue by directly estimating the "lack of knowledge" as the error made by the learner which empirically helps in better uncertainty estimates which lead to better performance on downstream tasks (like OOD Detection).

---

### Author Response · Authors · 2021-11-21
**Updated Draft**

We would like to thank the reviewers for all the thoughtful feedback!

Based on the feedback we have made several changes to the paper draft to address issues raised by the reviewers:
1. Improved discussion of Fig. 2, Section 2, and Section 3.1 to focus on the motivation for directly estimating uncertainty as the lack of knowledge and the challenges in interactive learning of this uncertainty
2. Clarified claims and contributions in the introduction to reflect the work more accurately
3. Added additional experimental results in the appendix: uncertainty estimation in a distribution shift setting, and added CI metrics in the drug combination experiments
4. Added additional discussion and references in the related work

We look forward to comments from the reviewers and further discussion!

---

> ### Comment · Reviewer_4Lh6 · 2021-11-22
> **Thanks for the updates**
>
> I thank the author for the updates and clarifications.
>
> Regarding section 3.2, in the new draft, the *second* argument made by the authors is very clear; If the main model is trained on new data, points that before should have been labeled as uncertain will no longer be uncertain. Thus, for the uncertainty predictor, it is as if like the target for some previously observed point changed. The authors increase the dimensionality of the input space in order for the uncertainty predictor to distinguish both scenarios.
>
> The *first* argument is still unclear to me unfortunately. I am not sure I understood it correctly even from the answer given to my review. My current understanding is that this paragraph is referring to bias introduced due to dataset shift in, for instance, an RL setting. To be specific, it is not clear to me why we would expect a sufficiently flexible variance-based method to fail here. I am also somewhat confused about how DEUP fixes the issue. Could the authors go into more detail about a specific example they have in mind when making these arguments?

---

> > ### Author Response · Authors · 2021-11-22
> > **Why do variance-based methods fail OOD**
> >
> > Thank you for the update and the interesting question regarding the different behaviors of DEUP and variance-based methods in distribution shift settings (such as RL or active learning).
> >
> > Here is an example that illustrates this difference.
> >
> > Consider the function shown in Figure 1 (top, in black). We can easily see that this function is made up of three parts: left (x < 0.5), middle (0.5 < x < 1.5), and right (x > 1.5). Assume we only have access to the left and the right parts of the training data (blue points) of Figure 1. The training distribution would have (-infinity, 0.5] U [1.5, infinity) as support.
> >
> > Now if we were to train an ensemble of large-enough neural networks, or a large-enough neural network with dropout, on this training data, we would end up with a quasi-constant mean prediction in [0.5, 1.5], close to zero, and a consistently low variance in this part of the input space (which is out-of-distribution).
> >
> > If an OOD point with “high error” is now incorporated to the training-set (e.g. x=0.7 y ~= 3), this would certainly change the mean prediction after retraining but would not alter the variance in this region significantly: there is no reason for the Ensemble or MC-Dropout learner to think “Oh ! This new OOD point is way beyond 3 sigma’s of what I predicted it to be; I should thus be more conservative and increase my uncertainty estimate for similar points”, let alone know what “similar” means.
> >
> > The same thing can be said when using Gaussian Processes. You can see in [this figure](https://ibb.co/MsrrMw4) how the variance almost doesn't change in the middle region after incorporating the point x=0.7 to the training set. There is no mechanism that allows the second GP to look at the first GP's variance in order to adjust its own variance estimates in that region.
> >
> > DEUP’s uncertainty estimator however can in principle extrapolate the fact that a large error was made at this x (the prediction was 0 whereas the true value is 3) to the other x’s of this middle region, simply because the secondary predictor would have been trained on the point ( phi(0.7), (3 - 0)^2), where phi(0.7) are the features describing the point x=0.7. This would lead to high uncertainty estimates in [0.5, 1.5], that get better as we include more OOD points. The extrapolation abilities are due to the usage of the features phi.

---

> > > ### Comment · Reviewer_4Lh6 · 2021-11-23
> > > **Thanks for the response.**
> > >
> > > The GP example helped clarify this for me as I think it is difficult to reason about this sort of thing with NNs. As I understand, the point is that  for existing methods the model choice determines the predictive variance to a large degree. I agree with this. The model choice is made using the data available during training and thus the errorbars are set expecting data with similar characteristics (e.g. level of smoothness) everywhere. If there is a change in data characteristics in some unseen region of input space, the errorbars will not be well calibrated.
> > >
> > > An interesting experiment to verify whether this is the case would be to successively increase the number of points observed by the GP (including them in the hyperparameter optimisation via marginal likelihood) or ensemble and in each step give the same new points to the DEUP error predictor. This would allow us to see the speed at which the errorbars of both approaches adapt to observing new points and if there really are gains to be made from considering the more free-form approach of DEUP.

---

> > > > ### Author Response · Authors · 2021-11-24
> > > > **Suggested Experiments**
> > > >
> > > > We thank the reviewer for the response and suggestions!
> > > >
> > > > The experiment you describe is precisely the premise of our experiments on Sequential Model Optimization (Section 5.2 and Appendix H). Here the model is trained on a few points initially and then in each round points are selected to be acquired based on their value as well as the uncertainty predicted by the model (using the expected improvement acquisition function). These points are then incorporated to train the model for acquisition in the next step. This is also the standard setting in Bayesian Optimization. As we demonstrate through our experiments DEUP can efficiently search the space (reach higher function values in fewer oracle calls), even in very high dimensional functions used for benchmarking Bayesian Optimization. This is only possible due to better uncertainty estimates from DEUP which can guide this search better.

---

> > > > > ### Comment · Reviewer_4Lh6 · 2021-11-24
> > > > > **Thanks for the response!**
> > > > >
> > > > > Thanks for the response. After the discussion I feel like key parts of the paper have become much clearer to me. I have raised my score accordingly. I would encourage the authors to clarify the first argument in section 3.2. I believe an illustration with the 1d GP case and multiple stages of learning, with different amounts of data, could be very helpful.

---

> > > > > > ### Author Response · Authors · 2021-11-24
> > > > > > **Thanks for discussion**
> > > > > >
> > > > > > Thank you for the discussion and valuable suggestions!
> > > > > >
> > > > > > We will update Section 3.2 with a more clear explanation in the camera-ready version, using the illustrations from the discussion, depending on the space constraints.
> > > > > >
> > > > > > We would be happy to discuss and address any further questions or comments on the paper!

---

### Decision · Program_Chairs · 2022-01-20

**Decision:**

Reject

**Comment:**

The manuscript develops a novel method for uncertainty prediction that can be used in the context of active or reinforcement learning problems. They consider experiments such as an OOD Detection task wherein a ResNet is trained on CIFAR10 and predictions must subsequently be made for in- versus out-of-distribution (SVHN) data.
The work develops an approach based on directly estimating epistemic (as opposed to a aleatoric) uncertainty by learning to predict generalization error and then subtracting estimated aleatoric uncertainty.
Reviewers found the essential approach to be novel and creative. However, there were several issues raised by reviewers that are not well addressed by responses by the authors. For example, Zaec worries about the dependence of the approach on an oracle for estimating aleatoric uncertainty. Multiple reviewers were concerned that this would make the approach unsuitable for many situations and thus limit the applicability of the ideas.
Multiple reviewers also found the manuscript to be difficult to understand. I agree with the sentiment. While there may indeed be an interesting and important idea here, the text and explication of the algorithm and approach are complicated and leave the reader unsure about the contribution. I would recommend that the authors invest time and effort in simplifying and streamlining the narrative and presenting the technical innovation so that it is easier to judge. In it's current form, the manuscript is premature for publication.